# A somatic role for the histone methyltransferase Setdb1 in endogenous retrovirus silencing

Masaki Kato[1], Keiko Takemoto[2] & Yoichi Shinkai[1]

Subsets of endogenous retroviruses (ERVs) are derepressed in mouse embryonic stem cells (mESCs) deficient for Setdb1, which catalyzes histone H3 lysine 9 trimethylation (H3K9me3). Most of those ERVs, including IAPs, remain silent if *Setdb1* is deleted in differentiated embryonic cells; however they are derepressed when deficient for *Dnmt1*, suggesting that Setdb1 is dispensable for ERV silencing in somatic cells. However, H3K9me3 enrichment on ERVs is maintained in differentiated cells and is mostly diminished in mouse embryonic fibroblasts (MEFs) lacking Setdb1. Here we find that distinctive sets of ERVs are reactivated in different types of *Setdb1*-deficient somatic cells, including the VL30-class of ERVs in MEFs, whose derepression is dependent on cell-type-specific transcription factors (TFs). These data suggest a more general role for Setdb1 in ERV silencing, which provides an additional layer of epigenetic silencing through the H3K9me3 modification.

[1] Cellular Memory Laboratory, Cluster for Pioneering Research, RIKEN, 2-1 Hirosawa, Wako, Saitama 351-0198, Japan. [2] Institute for Frontier Life and Medical Sciences, Kyoto University, 53 Shogoin, Sakyo, Kyoto 606-8507, Japan. These authors contributed equally: Masaki Kato, Keiko Takemoto. Correspondence and requests for materials should be addressed to M.K. (email: mkato@riken.jp) or to Y.S. (email: yshinkai@riken.jp)

Endogenous retroviruses (ERVs) are major components of mammalian genomes[1]. To silence ERV activity and transposition, hosts have evolved multiple silencing mechanisms to establish and maintain repressive heterochromatin formation across these elements[2]. We previously demonstrated that the H3K9me3 methyltransferase Setdb1 (Eset, KMT1E) represses intracisternal A particles (IAPs) and other ERVs in mouse embryonic stem cells (mESCs)[3,4]. Similarly, the depletion of Trim28 (Kap1, Tif1β), which is a co-repressor of Setdb1, leads to the derepression of IAP elements in mESCs[5]. Trim28 is recruited to ERVs via sequence-specific Krüppel-associated box zinc-finger proteins (KRAB-ZFPs), a large family of transcription factors (TFs) that might have expanded and co-evolved in mammalian genomes with the expansion of ERVs[6,7]. DNA methylation is mostly dispensable for the silencing of ERVs repressed by Setdb1 in early embryonic cells[3,4]. Setdb1 is also required for the silencing of ERVs at approximately embryonic day 13.5 (E13.5) in primordial germ cells[8], in which DNA is typically hypomethylated[9,10]. In contrast, in somatic and male germ-lineage cells, which are further differentiated, DNA methylation has been proposed to be the main mechanism to silence ERVs and non-long terminal repeat (LTR) retroelements. In this context, CpG dinucleotides of retroelements are densely methylated and *Dnmt1* inactivation leads to IAP derepression in embryos or differentiated mESCs upon depletion of LIF signals[11,12]. Thus, it is generally considered that the H3K9me3-mediated ERV silencing pathway in mESCs is rapidly replaced by a more permanent silencing mechanism, i.e., Trim28-mediated de novo DNA methylation, in differentiated embryonic cells[13–15]. Once DNA methylation is established, sequence-specific KRAB-ZFPs and Trim28 are no longer required[16,17].

DNA methylation patterns acquired during development have long been considered a stable epigenetic mark in somatic cells and adult cells. However, several recent studies have revealed that some ERVs are also derepressed in differentiated somatic cells lacking Trim28 or Setdb1[18–22]. In this study, we re-evaluate the role of Setdb1 in ERV silencing, not only in ESCs, but also in differentiated somatic cells, in which ERVs are heavily DNA methylated. We find that specific sets of ERVs are reactivated in different types of Setdb1-deficient somatic cells. Our data suggest that Setdb1 plays a more general role in ERV silencing, providing an additional silencing mechanism through H3K9me3.

## Results

**Derepression of distinct ERV families upon loss of Setdb1.** H3K9me3 enrichment in ERV family members has been detected in mESCs[3–5,23,24]. Although DNA methylation is important for ERV silencing in differentiated cells, the relevance of H3K9me3 marks is not well defined. To analyze whether the H3K9me3 marks on ERVs are important for silencing in differentiated cell types, we performed RNA sequencing (RNA-seq) analysis on *Setdb1* conditional knockout (cKO) immortalized mouse embryonic fibroblasts (iMEFs)[3], which is a model for differentiated cells. Our data were compared with previously published RNA-seq datasets for mESCs and other differentiated cell types with or without *Setdb1* KO[4,18,20]. The amount of Setdb1 in iMEFs is almost 10 times lower than that in ESCs, and depletion of Setdb1 by 4-hydroxytamoxifen (4OHT) in iMEFs induced growth defects, similar to that in *Setdb1* cKO mESCs. However, the growth recovered 8 days after 4OHT treatment in iMEFs was unlike that in ESCs[3] (Supplementary Fig. 1). We analyzed total RNA (rRNA was depleted) from untreated and 4OHT-treated *Setdb1* cKO iMEFs 5 days after treatment. An RNA-seq analysis of repeats in *Setdb1* cKO iMEFs revealed a substantially increased expression of ERVs after Setdb1 depletion, particularly five

elements annotated by Repbase, i.e., MMVL30-int, MuLV-int, RLTR4_Mm, RLTR4_Mm-int, and RLTR6_Mm (Fig. 1a, highlighted in red). In contrast, distinct ERV families were derepressed in other cell types when Setdb1 was removed[4,18–20,25]. For example, MMERVK10C exhibited the highest induction in *Setdb1* KO ESCs. Furthermore, IAPLTR1_, 1a_, and 2_Mm were highly derepressed in the fetal forebrains of *Setdb1* KO mice, and RLTR3_Mm was specifically induced in *Setdb1* KO granulocyte/macrophage progenitors (GMPs) or bone marrow Lin⁻ Sca-1⁺ c-Kit⁺ (LSK) cells.

**A viral defense response is induced in *Setdb1* KO iMEFs.** We also examined the transcription of non-repeats, and identified 244 RefSeq genes that were upregulated by at least 2-fold in 4OHT-treated *Setdb1* cKO iMEFs, compared to untreated cKO iMEFs (Supplementary Fig. 2a and Supplementary Data 1). A Gene Ontology analysis of the upregulated genes revealed the major representation of genes involved in immune or defense responses (Supplementary Fig. 2b). In particular, we observed the upregulation of several genes related to interferon (IFN) responses, such as *Irf7, Irf9, Usp18,* and *Stat1* in *Setdb1* KO iMEFs (Supplementary Fig. 2c). *Irf7*, which is a master regulator of type-I IFN-dependent immune responses, was upregulated by more than 20-fold.

To determine how many genes upregulated in *Setdb1* KO iMEFs were directly repressed by Setdb1, we performed a chromatin immunoprecipitation sequencing (ChIP-seq) analysis of H3K9me3 in wild-type (WT) iMEFs. Surprisingly, only 1% of the promoter regions of genes that were upregulated in *Setdb1 KO* iMEFs (2/244) were marked by H3K9me3 (Supplementary Fig. 3a), indicating that almost all genes induced by *Setdb1* KO are indirectly regulated by Setdb1. This is consistent with the results obtained for *Setdb1* cKO mESCs[4]. Derepressed ERVs might play a role in altering the gene expression profile of *Setdb1* KO cells, since transcriptionally reactivated ERVs significantly enhance the transcription of their neighboring genes[4,18]. Thus, we analyzed the correlation between upregulated genes in *Setdb1* KO iMEF cells and ERV insertions. In MEFs, ~50% of the H3K9me3-marked upregulated genes (11 out of 27 genes) had ERV or LINE insertions with H3K9me3 enrichment (Supplementary Fig. 3a). H3K9me3 on these retroelements diminished in MEFs after Setdb1 depletion (Supplementary Fig. 3b and Supplementary Data 2). However, we did not observe any derepressed elements inserted in the upregulated genes in *Setdb1* KO iMEFs, unlike those seen in *Setdb1* KO ESCs or forebrain cells[4,18]. Furthermore, we calculated the ratio between upregulated genes with and without ERV insertions, and compared it with those for the downregulated and unchanged genes. There was no statistically significant difference between the values for the upregulated genes and those for the other genes (the ratio of upregulated genes is 0.0472, downregulated 144 genes 0.0434, and unchanged 22533 genes 0.0467, respectively, thus ERVs seen at upregulated genes were expected by chance (P = 0.12, binomial test)).

**H3K9me3 on ERVs are preserved in distinct cell types.** Next, we tried to determine whether H3K9me3 marks on ERVs seen in ESCs are preserved in distinct differentiated somatic cells. For this, we utilized published H3K9 ChIP-seq datasets for ESCs[24], fetal forebrain cells[18], iMEFs[24], and GMPs[20], and generated NGS plots using the positions of H3K9me3 peaks for representative ERVs (MMERVK10C-int, IAPEz-int, RLTR4_Mm-int, RLTR6-int, and MMVL30-int), which are all derepressed in at least one of the studied *Setdb1* KO cell types (Fig. 1b). In this analysis, we selected full-length ERV elements for each family. Comparison of

H3K9me3 profiles demonstrated that the pattern of H3K9me3 peaks on ERVs are chiefly conserved among these distinct cell types, although H3K9me3 enrichment seems lower in differentiated cell types than in ESCs, which is consistent with the results from a previous report[24].

**Setdb1-dependent H3K9me3 marks on ERVs**. To address why Setdb1 KO led to the derepression of distinct sets of ERV families, we performed ChIP-seq analysis of H3K9me3 on Setdb1 cKO iMEFs treated with or without 4OHT (KO) for 7 days, and compared the data for the WT and KO samples. Genome-wide

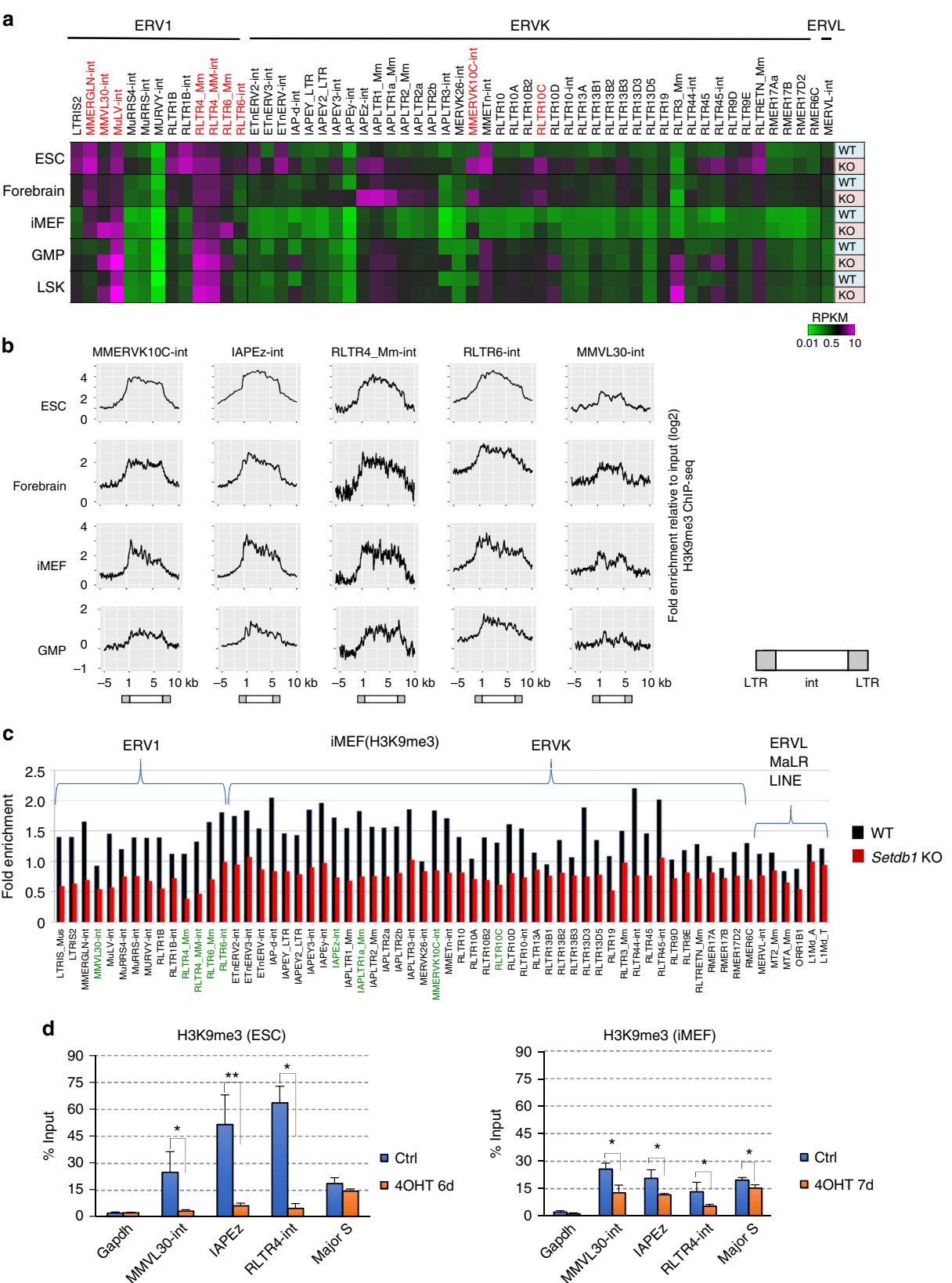

ChIP-seq analysis revealed that most of the enriched H3K9me3 on the examined ERVs was reduced in *Setdb1* cKO iMEFs due to 4OHT treatment; however, the reduction level on each ERV was variable (Fig. 1c and Supplementary Data 2). We observed a slight reduction of H3K9me3 in LINE families (Fig. 1c and Supplementary Fig. 3c). To validate the ChIP-seq data and to compare Setdb1-dependent H3K9me3 on ERVs between iMEFs and ESCs, we performed a ChIP-qPCR analysis of H3K9me3 on selected ERVs and major satellite repeats for both cell types (Fig. 1d). Consistent with previous results and NGS plot data (Fig. 1b), H3K9me3 was enriched in the repeat domains of WT ESCs, but exhibited much less enrichment on such repeats in iMEFs. After Setdb1 depletion in *Setdb1* cKO mESCs and iMEFs, we observed a decrease in H3K9me3 enrichment at the MMVL30-int, IAPEz, and RLTR4-int loci, suggesting that those H3K9me3 modifications were induced by Setdb1 in iMEFs (Fig. 1d). In contrast, we did not observe any significant reduction of H3K9me3 at major satellite loci in ESCs; however, a slight reduction was observed in iMEFs, where Suv39h is responsible for most of H3K9me3[24]. Thus, Setdb1 regulated H3K9me3 marks at many of the ERV loci that were examined.

**Derepression of the VL30 family in Setdb1 KO MEFs**. RNA-seq analysis of repeats in *Setdb1* cKO iMEFs revealed a substantial increase in the expression of MMVL30-int and RLTR6_Mm sequences (Fig. 1a). RLTR6_Mm is an LTR element flanking MMVL30-int in the VL30 family (virus-like 30, MMVL30). The VL30 ERV family can be largely classified into two groups, based on their internal sequences. One group has RLTR6-int elements, which are predominantly intact ERVs with coding regions for retroviral Gag, Pol, and Env proteins; the second group has MMVL30-int elements, which are likely derived from RLTR6-int elements, but have lost their coding regions (Fig. 2a). In addition, VL30-containing RLTR6 LTRs can be divided into four subgroups on the basis of their LTR U3 sequences[26]. RLTR6-int type VL30 is mostly flanked by U3 class II RLTR6s; therefore, we called it VL30 U3 class II. MMVL30-int is flanked by U3 class I, III, or IV RLTR6s, and were thus called VL30 U3 class I, III, or IV, respectively (Fig. 2a). U3 class III RLTR6s contain a DR2-type all-trans retinoic acid (atRA)-response element (RARE), and U3 class III VL30 expression is induced by atRA in keratinocytes[27]. On the other hand, U3 class II RLTR6s contain DR2-type and DR5-type RAREs (Fig. 2a).

To identify which VL30 U3 classes are repressed by Setdb1 in iMEFs, we analyzed the RNA-seq data of iMEFs with and without Setdb1 depletion by treatment with 4OHT. After 4OHT treatment, VL30 U3 class I and III were significantly derepressed by more than 10-fold and 5-fold, respectively, and class IV was marginally induced, but class II was unaffected (Fig. 2b). We generated NGS plots, using the positions of H3K9me3 peaks on

VL30 U3 class I, III, and VI elements containing the MMVL30-int sequence (Fig. 2c). They shared similar H3K9me3 peak patterns in the three cell types (ESCs, forebrain cells, and iMEFs), even though significant derepression of VL30 U3 class I and III was only observed in *Setdb1* KO iMEFs. We also observed similar H3K9me3 peak patterns on U3 class II-containing RLTR6-int sequences in the three cell types (Fig. 1b). ChIP-seq analysis of *Setdb1* KO iMEFs confirmed the loss of H3K9me3 in all four classes of VL30 (Fig. 2d).

**Distinct regulation mechanism for each VL30 U3 class**. Since specific U3 classes of VL30 were reactivated by *Setdb1* KO in iMEFs, we expected that each class would have different regulation mechanisms for transcription. VL30 U3 class II has a predominantly intact internal sequence (Repbase annotation: RLTR6-int) with coding regions for retroviral Gag, Pol, and Env proteins (Fig. 2a). The H3K9me3 peaks are enriched not only on the 5′ LTR, but also across the internal region (Fig. 2d right). As stated, this VL30 U3 class II expression is restricted to Rar α activity on LTR RAREs in the liver[28]. Therefore, the lack of derepression of VL30 U3 class II in *Setdb1* KO iMEFs might be due to an absence of endogenous atRA in MEFs. We then evaluated the requirement of atRA for VL30 U3 class II transcription. The addition of atRA alone did not activate VL30 U3 class II expression in *Setdb1* cKO iMEFs, but substantial induction was observed after both atRA and 4OHT treatments (Fig. 3a). In mESCs, *Setdb1* KO or atRA treatment alone weakly activated VL30 U3 class II; this activation was further enhanced after both atRA and 4OHT treatment (Fig. 3b). Thus, for the U3 class II VL30 expression in both MEFs and mESCs to be maximum, a loss of H3K9me3 and the presence of atRA is required. We also examined the effect of atRA on VL30 class I copies, but impact of atRA addition to *Setdb1* KO is marginal or absent for iMEFs and ESCs, respectively.

Next, we focused on distinct individual elements of VL30 U3 class I (total 71 elements) to further investigate interactions between Setdb1-mediated H3K9me3 and transcriptional silencing. Twenty-two elements have a typical proline tRNA primer-binding site (PBS-pro) near the 3′ end of their 5′ LTR. Not all of these are derepressed in *Setdb1* KO iMEFs (Fig. 3c top); PBS-pro-negative elements are relatively more derepressed than their PBS-pro-positive counterparts. Although unique mapping was incomplete due to highly homologous sequences, we tried to analyze the reduction of H3K9me3 at the 5′ LTR regions of 71 VL30 U3 class I elements in *Setdb1* KO iMEFs using ChIP-seq data. We observed a significant reduction of H3K9me3 at most of those distinct element loci, but the reduction rate varied (Fig. 3c bottom). We observed a marginal correlation between the derepression rate and reduction rate of H3K9me3 for those VL30 class I elements in *Setdb1* KO iMEFs (Supplementary

**Fig. 1** Different ERV families are derepressed by *Setdb1* KO in different cell types. **a** Cell-type-specific ERV derepression in *Setdb1* cKO cells. Expression of ERV families in *Setdb1* cKO ESC (day 6 after treatment with 4OHT (KO) or no treatment (WT))[4], iMEF (day 5 after treatment with 4OHT (KO) or no treatment (WT)), and E14.5 forebrain cells from WT and *Emx2*-Cre:*Setdb1* fl/fl mice (KO)[18]. For GMP and LSK cells, bone marrow cells from *Rosa*-CreERT: *Setdb1* cKO mice were transplanted into irradiated recipient mice, GMP and LSK cells were then isolated after injection of 4OHT for 2 weeks (KO) or control injection (WT)[20]. Only ERVs derepressed (≥2 fold) in at least one of the analyzed cell types with *Setdb1* KO are listed. Heatmap indicates the relative expression level of representative ERV families (the RPKM value). The ERVs derepressed (≥1.5 fold) in *Setdb1* KO iMEFs are highlighted in red. **b** H3K9me3 intensity profiles on different ERV families in different cell types. NGS plots show the fold enrichment of H3K9me3 from −5 kb to 10 kb around genomic ERV elements in ESC, forebrain, iMEF, and GMP. We selected ERVs containing -int element with flanked LTRs (See Methods). Position 1 is 5′ start site of the -int element. Positions of LTRs and int for each ERV element are indicated below the plots. **c** Bar plots showing the loss of H3K9me3 in ERV families in *Setdb1* KO (day 7 after 4OHT treatment: red bar) vs. WT (black bar) iMEF. The *y* axis indicates the fold enrichment of normalized ChIP read density relative to input. ERV names written in green are analyzed in Fig. 1b. **d** ChIP-qPCR of H3K9me3 in the *Gapdh* promoter region indicated ERVs and major satellite (Major S) loci of *Setdb1* cKO ESCs and iMEFs. Values are means ± s.d. from independent experiments (*n* = 3). *0.005 < *P* < 0.05, **0.0005 < *P* < 0.005, Student's *t*-test

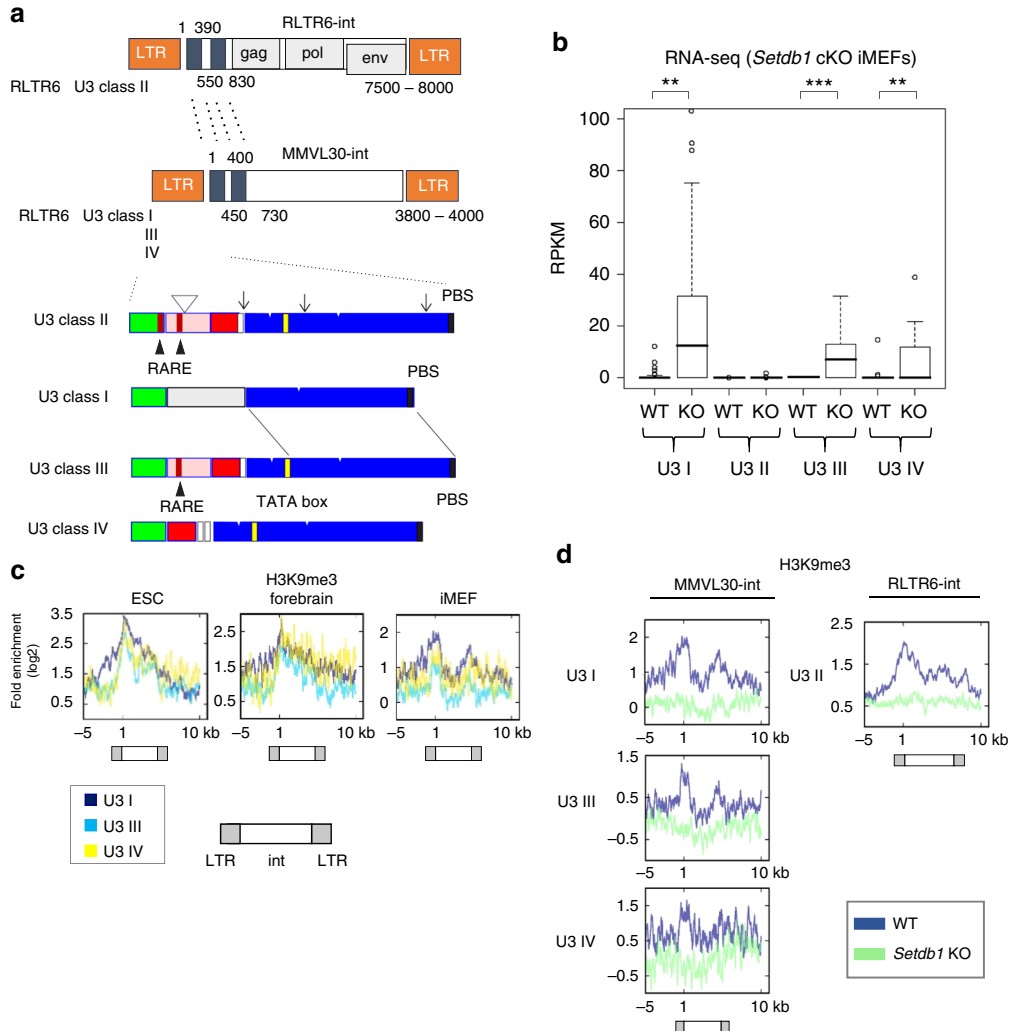

**Fig. 2** MMVL30 families are derepressed in *Setdb1* KO iMEFs. **a** Schematic structures of VL30 families. Blue boxes indicate highly homologous sequences between MMVL30-int and RLTR6-int. High magnification of LTR regions of U3 classes. Color boxes indicate highly homologous sequence among LTRs. White arrowhead and arrow indicate unique insertion and mutation in class II, respectively. Black arrowhead indicates RARE sites. **b** Expression of VL30 U3 classes in *Setdb1* KO iMEFs. Boxplots show that expression of VL30 U3 classes in *Setdb1* KO iMEFs (day 5 after 4OHT treatment (KO) or no treatment (WT)). RNA-seq reads overlapping each VL30 loci are counted and normalized using the length of the locus and million mapped reads (RPKM). U3 I (71 loci), U3 II (189 loci), U3 III (66 loci), and U3 IV (26 loci). **0.0005 < P < 0.005, ***P < 0.0005, Student's *t*-test. **c** NGS plots of H3K9me3 ChIP-seq data at genomic VL30 loci (RLTR6_Mm - MMVL30-int) of ESCs, E14.5 forebrain cells and iMEFs. MMVL30-int with RLTR6_Mm U3 I (36), with U3 III (40) and with U3 IV (10). The number inside of parenthesis indicates the number of individual each element analyzed. LTRs are aligned and MMVL30-int starts from position 1. H3K9me3 intensity profiles are shown with dark blue line (U3 I), light blue one (U3 III), and yellow one (U3 IV). Positions of LTRs and int are indicated below the plots. **d** NGS plots of H3K9me3 ChIP-seq data at genomic "RLTR6_Mm - MMVL30-int" and "RLTR6_Mm - RLTR6-int" loci in *Setdb1* cKO iMEFs (day7 after 4OHT treatment (KO) and no treatment (WT)). H3K9me3 fold enrichments (log2) are shown with dark blue line (WT) and light blue one (KO)

Fig. 4). Since only a subset of VL30 U3 class I was derepressed in *Setdb1* KO cells, this derepression variation cannot be explained just by the reduction rate of H3K9me3.

The structural analysis of LTRs of VL30 has been reported to show a possible requirement of tissue-specific TFs[29]. Thus, we focused on the potential role of TFs in VL30 U3 class I derepression in *Setdb1* KO iMEFs. Multiple DNA sequence alignment of the LTRs for VL30 U3 class I, generated using Clustal Omega, revealed that the silent populations differ slightly at the DNA sequence level (Fig. 3d). A motif analysis using the Patch 1.0 database predicted binding sites for several TFs, including AP-1, Ets1, and Elk, in the LTRs for VL30 U3 class I, but motifs for Elk family and Ets1-binding sites were mutated in the silent or low derepressed populations (Fig. 3d and Supplementary Data 3). Elk-1 is a member of the Ets family of

TFs. Elk-1 and Ets1 appear to be direct targets of activated MAP kinase (MAPK)[30]. Thus, we examined the requirement of MAPK activity for VL30 U3 class I derepression in *Setdb1* KO iMEFs. PD0325901 is a potent MEK inhibitor that suppresses the phosphorylation of ERK[31]. We treated *Setdb1* cKO iMEFs with PD0325901 for the final 24 h of a 5-day treatment with 4OHT and observed a strong inhibition of VL30 derepression in PD0325901-treated iMEFs (Fig. 3e). RNA-seq analysis showed that induction of low derepressed VL30 elements with mutated Elk/Ets binding sites, such as #58 and #47 copies were also diminished by the MEK inhibitor treatment as similar to those containing intact Elk/Ets binding sites (Supplementary Fig. 5a). Indeed, AP-1 is known to be present downstream of the MAPK pathway to activate VL30 elements[32,33] and AP-1 binding site is mostly intact in the derepressed VL30 elements regardless of Elk/

Ets binding site mutation(s) (Fig. 3d and Supplementary Data 3). Thus, we speculate that AP-1 also contribute to activation of VL30 U3 class I elements after Setdb1 depletion. To test the direct involvement of Ets1 in VL30 derepression, we performed ChIP-qPCR analysis of Flag-tagged Ets1 to see an enrichment of Ets1 at VL30 loci. We observed a slight enrichment of Ets1 at U3 class I #9 loci (Ets binding site is intact) after Setdb1 depletion, but no enrichment at #6 and #18 copies, in which the Ets-binding site is mutated (Supplementary Fig. 5b), suggesting that Ets1 might contribute to VL30 derepression. However, the Ets family is large, and we do not rule out the possibility that other Ets proteins are involved in VL30 derepression. Considering these results, we conclude that most VL30-class ERVs are silenced by the Setdb1 pathway in MEFs. However, only specific sets of ERVs are potentially active, depending on whether the regulatory elements in their LTRs are compatible with the activity of TFs in distinct cell types.

**Distinct requirements for epigenetic marks in ERV silencing.** DNA methylation has an important role in proviral silencing in somatic cells[12,34], but the impact of DNA methylation over entire

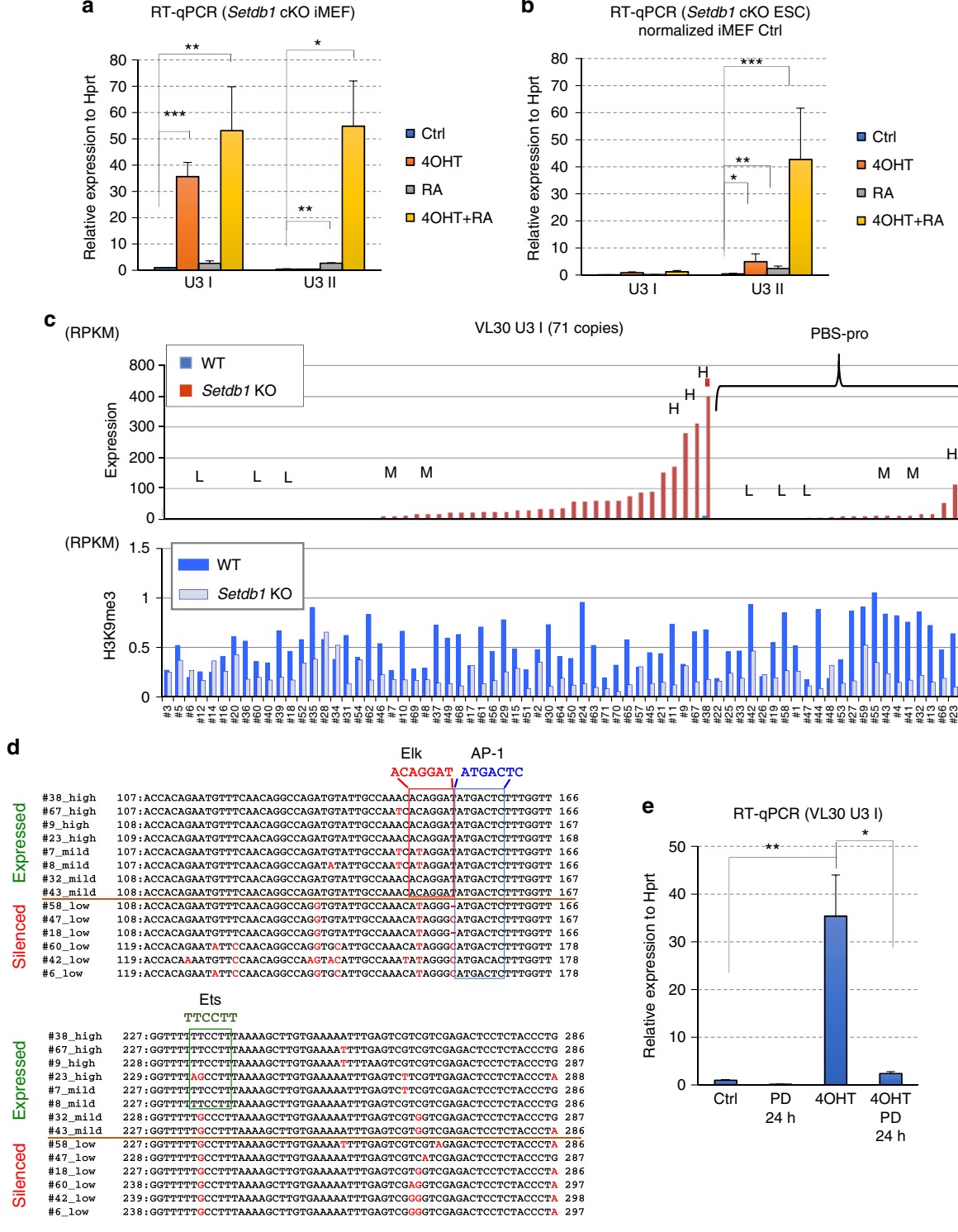

ERVs has not been well characterized, especially in differentiated cells. Therefore, we performed RNA-seq analysis on *Dnmt1* knock down (KD) iMEFs using siRNA. Surprisingly, only a small fraction of ERV families was derepressed in *Dnmt1* KD iMEFs, involving mostly the IAPEz family and the flanking LTR (IAPLTR1_Mm), with a much weaker fold induction of MuLV-int and MMVL30-int (Fig. 4a top). We also reanalyzed public RNA-seq data for WT and *Dnmt1*[−/−] iMEFs in the *p53*[−/−] background[35] to examine the derepression of ERVs. Consistent with the *Dnmt1* KD result, we observed a prominent derepression of the IAPEz family and the flanking LTR (IAPLTR1_Mm) (Supplementary Fig. 6). In *Dnmt1* KO ESCs, the ERVs IAPEz and IAPLTR1_Mm were also prominently derepressed (Fig. 4a bottom)[36].

Since DNA methylation is involved in silencing a subset of ERVs, such as IAPEz, in mESCs in the absence of H3K9me3[4], we examined whether the simultaneous depletion of DNA methylation and Setdb1 leads to a higher level of ERV reactivation in iMEFs than in the *Setdb1* KO alone. *Setdb1* cKO iMEFs were treated with siRNA specific to *Dnmt1* alone or in combination with 4OHT, and then RNA-seq analysis was performed. We did not observe a strong augmentation of ERV derepression by the simultaneous depletion of Dnmt1 and Setdb1 (Fig. 4a top). In contrast to VL30, IAPs possess the highest CpG density[36]. We also confirmed the RNA-seq data using RT-qPCR. Consistent with the *Dnmt1* KD iMEF and *Dnmt1*[−/−] iMEF RNA-seq results, *Dnmt1* KD alone induced a strong reactivation of IAPEz in iMEFs (Fig. 4b), but the *Setdb1* KO alone did not show any reactivation. IAPEz did not show increased reactivation by simultaneous depletion of Dnmt1 and Setdb1. On the other hand, we observed a slight derepression of VL30 by *Dnmt1* KD alone. In addition, VL30 showed a minor augmentation in reactivation following the simultaneous depletion of Dnmt1 and Setdb1, compared to that by *Setdb1* KO alone (Fig. 4b). For some of ERVs like RLTR4, we observed a mild derepression and marginal augmentation in induction due to the simultaneous depletion of both Setdb1 and Dnmt1 (Fig. 4b).

We further analyzed distinct individual elements of VL30 U3 class I to investigate the role of DNA methylation and H3K9me3 in their silencing (Fig. 4c). Some of the Setdb1-repressed elements like #9 and #38 (all non-PBS-pro) were also derepressed by *Dnmt1* KD with Dnmt1-specific siRNA, but most of these U3 class I elements were not derepressed, and the impact of the simultaneous depletion of Dnmt1 and Setdb1 was marginal (Fig. 4c). *Dnmt1* siRNA treatment significantly reduced CG methylation levels on IAPEz (at the Mnd1 locus) and VL30 (PBS-pro) (Supplementary Fig. 7a); however, we did not detect clear reduction in DNA methylation at the analyzed VL30 5′ LTR sequences (individual locus of non-PBS-pro copies, #21, #38 and pool of PBS-pro elements loci) after the depletion of Setdb1

(Supplementary Fig. 7b and 8). Collectively, the silencing of VL30 in MEFs mostly depends on the Setdb1 pathway, even though DNA methylation contributes to this phenomenon to some extent. In contrast, the Dnmt1-mediated pathway is critical for the repression of IAPEz in iMEFs. Both the Dnmt1 and Setdb1 pathways contribute to IAPEz silencing in mESCs[4].

**Long-term-cultured *Setdb1* KO iMEFs.** In our RNA-seq analysis using long-term-cultured *Setdb1* KO iMEFs, we noticed that reactivated ERV expression was lower than that in 4OHT-treated *Setdb1* cKO iMEFs (Supplementary Fig. 9a). To further examine the dynamics of VL30 U3 class I derepression after the acute depletion of Setdb1, we performed RT-qPCR analysis of VL30 U3 class I from day 0 to 11 days after the addition of 4OHT (Supplementary Fig. 9b). Upregulation of VL30 U3 class I was observed from day 4, and by day 8, the transcription level was more than 60-fold higher than the WT levels. However, this VL30 U3 class I expression decreased progressively after day 9. We performed ChIP-qPCR for H3K9me3 in WT, 4OHT-treated *Setdb1* cKO iMEFs, and long-term-cultured *Setdb1* KO iMEFs, and observed a lower enrichment of H3K9me3 at VL30 loci in long-term-cultured *Setdb1* KO iMEFs, which was not the case in WT cells (Supplementary Fig. 9c). We did not observe any recovery of H3K9me3 after long-term culturing. It is possible that other epigenetic repressive modifications, like DNA methylation or H3K27me3 modification, function to silence VL30, as backup mechanisms. Additional treatment of long-term-cultured *Setdb1* KO iMEFs with 5-Aza-dC, a Dnmt inhibitor, did not increase VL30 expression (Supplementary Fig. 9d). Thus, the desensitization of VL30 expression in long-term-cultured *Setdb1* KO iMEFs is not due to DNA methylation as a backup mechanism. To examine the potential role of H3K27me3, we treated cells with GSK126, which is a specific inhibitor of Ezh2[37]. Depletion of H3K27me3 in long-term-cultured *Setdb1* KO iMEFs did not augment the expression of VL30 induced by *Setdb1* KO alone; however, the overall H3K27me3 level was significantly reduced (Supplementary Fig. 9e, f).

**Distinct requirements for Setdb1, Trim28, and Zfp809.** Zfp809, a member of the KRAB-ZFP family, initiates the silencing of VL30 family members in a sequence-specific manner via the recruitment of Trim28–Setdb1 complexes[38]. Zfp809 binds to PBS-pro, which is used by some retroviruses to prime reverse transcription. Thus, Zfp809 can recruit Trim28 and Setdb1 to ERVs possessing PBS-pro[39–41]. We analyzed the RNA-seq data of *Setdb1* KO iMEFs and *Zfp809* KO MEFs[38] to compare non-PBS-pro and PBS-pro, with respect to VL30 derepression. The *Zfp809* KO MEFs were obtained from the KO embryo[38]. We observed a significant derepression of the non-PBS-pro VL30 group in

**Fig. 3** VL30 activation requires cell-type-specific TFs. **a**, **b** RT-qPCR of the VL30 U3 classes in RNA from *Setdb1* cKO iMEFs (**a**) or ESCs (33#6,[3]) (**b**), untreated or treated with 1 μM all-trans retinoic acid (atRA), with or without 4OHT (*n* = 3 biological replicates). For ESC data (*n* = 2 biological replicates), we normalized each expression to VL30 U3 class I LTR expression of control iMEF. Error bars represent s.d. *0.005 < *P* < 0.05, **0.0005 < *P* < 0.005, ***P* < 0.0005, Student's *t*-test. **c** Upper panel; the derepression induced by Setdb1 depletion in each VL30 U3 class I element (totally 71 elements) is shown. 22 loci on the right side have typical PBS-pro near the 3′ end of 5′ LTR. Selected loci for parallel alignment shown in **d** are denoted by H, M, and L symbols based on their derepression (H: high assigned to the locus where RPKM[KO] (RPKM of *Setdb1* KO) is >100, M: mild assigned to 10 ≤ RPKM[KO] < 100, and L: low assigned to RPKM[KO] < 10, respectively). Lower panel: H3K9me3 ChIP-seq read counts on 5′ LTR at each VL30 U3 class I locus in *Setdb1* cKO iMEFs (no treatment (WT: blue bar) and day 7 after treatment of 4OHT (KO: light blue bar)) are shown. **d** Alignment of U3 sequences of VL30 U3 class I. The red box indicates the consensus sequence of Elk family binding sites. The blue and green boxes indicate the consensus sequences of AP-1 and Ets, respectively. High, mild, and low indicate in Fig. 3c upper panel. Entire alignment of VL30 class I shown in Fig. 3c is in Supplementary Data 3. We defined the copies for RPKM[KO] < 10 as silenced. **e** Inhibition of the MAPK pathway precludes the transcriptional activation of VL30 in *Setdb1* KO iMEFs. RT-qPCR analysis of VL30 U3 class I. 4OHT treatment was applied for the first 4 days and cells were harvested at day 6. The MEK inhibitor PD0325901 (PD; 1 μM) was added for the last 24 h (*n* = 2 biological replicates). Error bars represent s.d. *0.005 < *P* < 0.05, **0.0005 < *P* < 0.005, Student's *t*-test

*Setdb1* KO MEFs, but not in *Zfp809* KO MEFs (Fig. 5a). In the PBS-pro VL30 group, both *Setdb1* and *Zfp809* KO MEFs showed derepression of VL30 (Fig. 5a); however, derepression in the *Setdb1* KO was less prominent than that in the non-PBS-pro group, as shown in Fig. 3c. These results indicate that ERVs possessing non-PBS-pro escape repression by Zfp809, but are targeted by other silencing factors that can recruit Setdb1. Zfp809 is required to initiate stable epigenetic silencing during development, but not to maintain silencing in somatic cells[38].

To confirm whether Trim28 is also required for the maintenance of ERV silencing, previously published RNA-seq data from *Trim28* KO MEFs, in which Trim28 was depleted in

the conditional KO MEFs[5], were compared with our *Setdb1* KO iMEF data. In addition to the lack of increase in IAPEz expression, which was published previously, an increase of less than 2-fold in VL30 (MMVL30-int) expression in *Trim28* KO MEFs was observed (Fig. 5b). It is possible that the derepression of ERVs is desensitized after long-term culture of *Trim28* KO MEFs. Therefore, we utilized CRISPR-gRNA systems to inactivate *Trim28*, and analyzed the silencing of VL30 in iMEFs in a short time period. We also used the CRISPR-gRNA system for *Setdb1* inactivation. Both *Trim28* and *Setdb1* gRNAs efficiently diminished protein expression of their target molecules, although the cells were mixture of WT, and the incomplete and complete KO populations

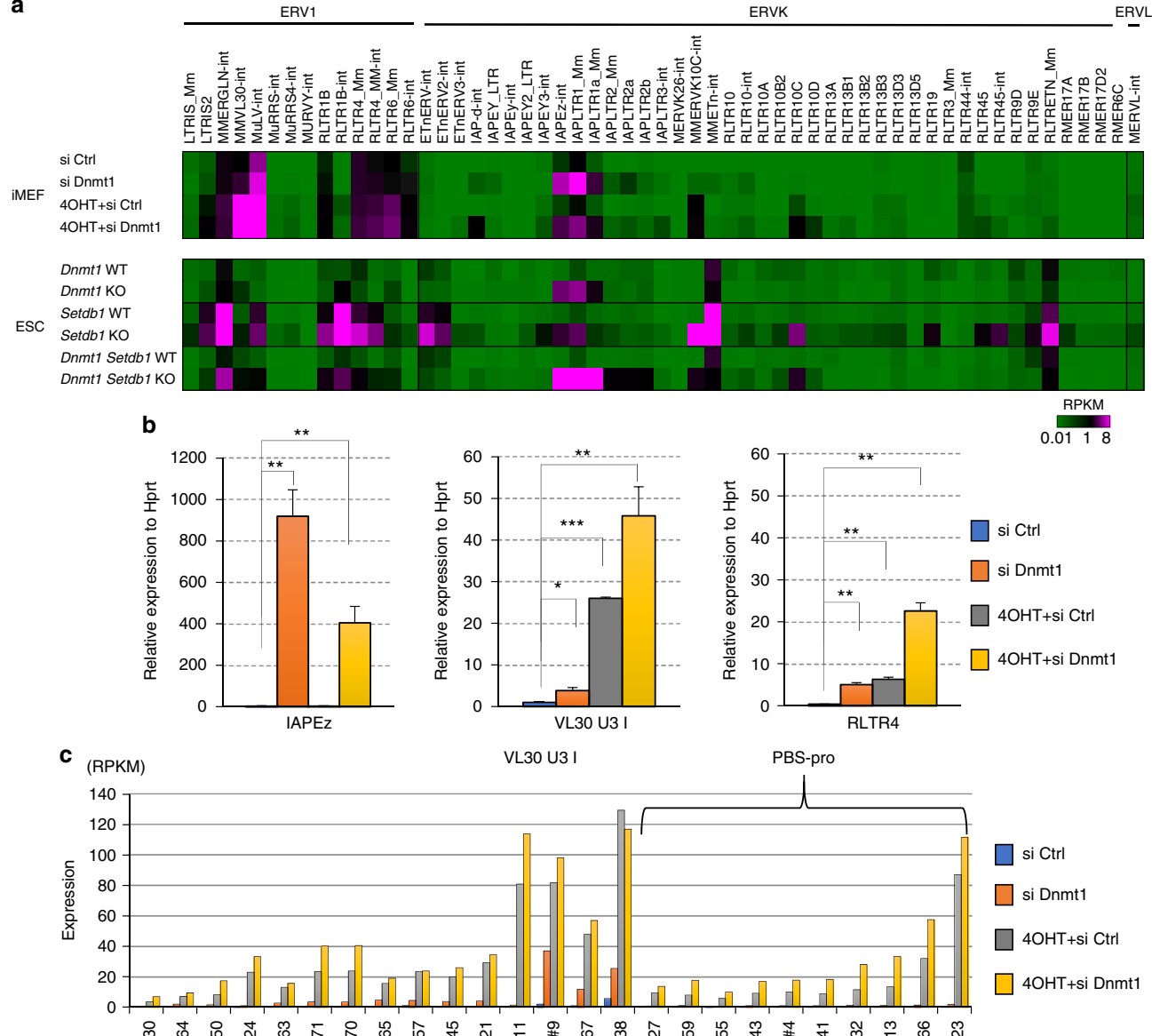

**Fig. 4** Dnmt1 and Setdb1 are required for the silencing of distinct sets of ERVs. **a** Expression of ERV families by the loss of Dnmt1, Setdb1, and simultaneous depletion of Dnmt1 and Setdb1 in iMEFs and ESCs. To obtain mRNA-seq data of iMEFs, *Setdb1* cKO iMEFs were transfected with siRNA against *Dnmt1* or control siRNA in combination with or without Setdb1 depletion by the 4OHT treatment, then mRNA was isolated. RNA-seq data from *Dnmt1* cKO or double cKO of *Dnmt1* and *Setdb1* ESCs (6 days after treatment of 4OHT or without treatment)[36] were reanalyzed. Heatmap indicates the magnitude of the RPKM value. **b** *Setdb1* cKO iMEFs were transfected with siRNA against *Dnmt1* either alone or in combination with the loss of Setdb1. RT-qPCR of VL30 U3 class I RLTR6, IAPEz, and RLTR4 was performed. Values represent mean expression relative to VL30 U3 I control ($n = 2$ biological replicates). Error bars represent s.d. *$0.005 < P < 0.05$, **$0.0005 < P < 0.005$, ***$P < 0.0005$, Student's *t*-test. **c** The derepression induced by *Dnmt1* KD, *Setdb1* KO, and simultaneous depletion of Dnmt1 and Setdb1 in iMEFs in each VL30 U3 class I element (totally 71 elements) is shown. The number of RNA-seq reads overlapping with a locus was divided by the length of the locus and normalized by million mapped reads (RPKM) as shown in Fig. 3c top

(Fig. 5c, right panel). The induction of U3 class I VL30 expression in iMEFs by this *Setdb1* gRNA treatment was greater than 30-fold induction. In contrast, we observed an ~2-fold increase in VL30 expression using *Trim28* gRNA (Fig. 5c, left panel), consistent with the results for the *Trim28* KO MEFs (Fig. 5b). Thus, these results indicate that Trim28 is mostly dispensable for the maintenance of VL30 silencing in MEFs, like Zfp809.

Finally, to test whether restoring Setdb1 expression can reverse U3 class I VL30 expression in long-term-cultured *Setdb1* KO iMEFs, we stably transfected cells with a transposon-based Setdb1 expression vector. Exogenous Setdb1 was capable of repressing VL30 upregulation, but this expression was still higher (~3-fold) than that in WT iMEFs (Fig. 5d). These results suggest that Setdb1 is recruited to VL30 loci via additional, unknown factors to initiate silencing in iMEFs.

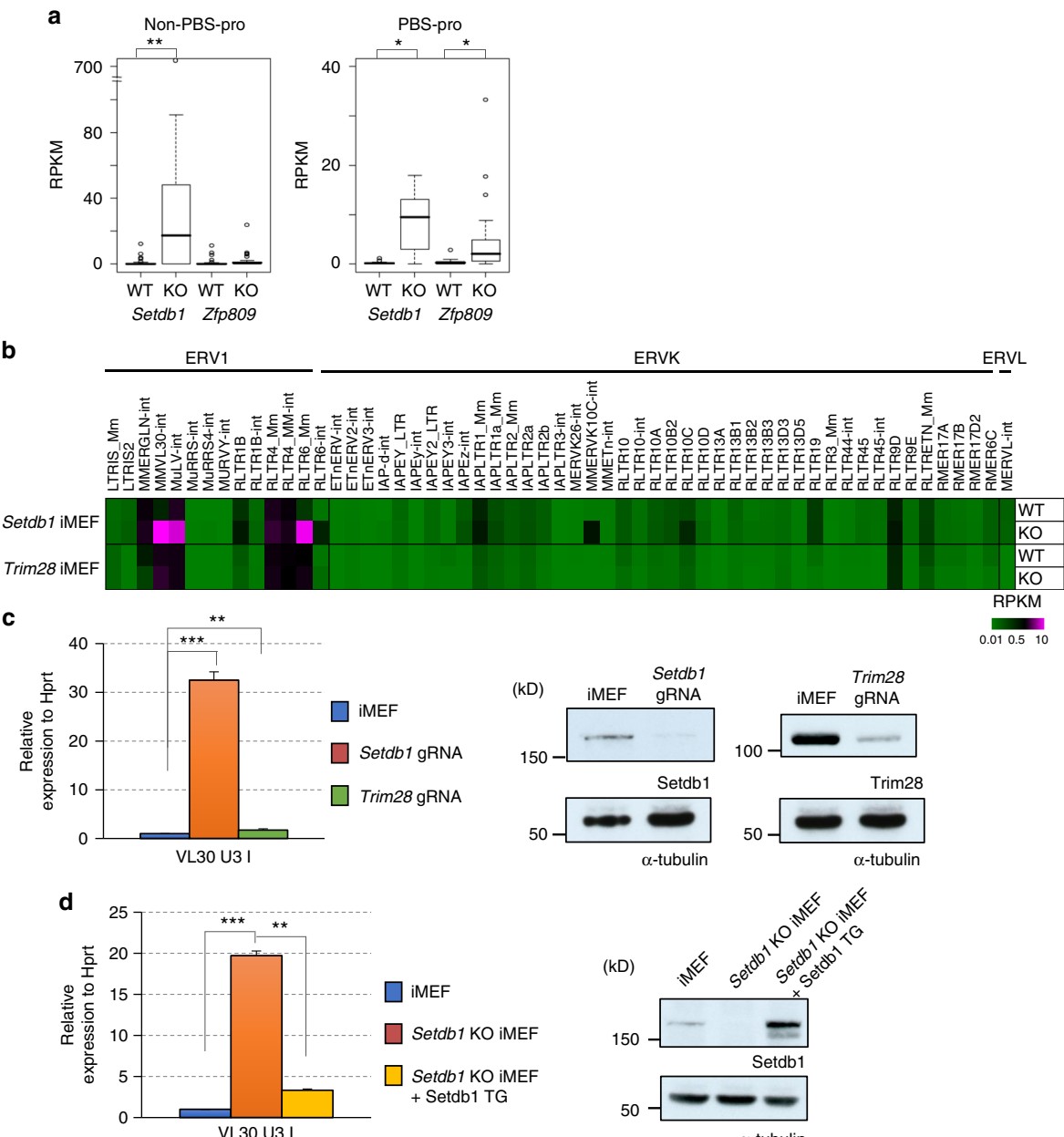

**Fig. 5** Distinct requirement of Setdb1, Trim28, and Zfp809 for VL30 silencing. **a** Boxplots indicate silencing effects on VL30 U3 I by Setdb1 and Zfp809. 71 VL30 U3 I (MMVL30-int with RLTR6_Mm U3 I) are divided into two groups according to with (22 loci) or without (49 loci) PBS-pro sequences. RNA-seq data of *Setdb1* cKO iMEFs (no treatment (WT) or 4OHT 5d (KO)), *Zfp809* KO MEFs derived from mutant embryos (KO) and wild type (WT)[38] are shown. *0.005 < P < 0.05, **0.0005 < P < 0.005, Student's *t*-test **b** Expression of ERV families in *Trim28* KO MEFs. RNA-seq data for *Trim28* WT and KO MEFs[5] was reanalyzed. Heatmap indicates the magnitude of the RPKM value. For comparison, *Setdb1* cKO iMEF data (Fig. 1a) was shown above. **c** *Trim28* KO only marginally lead to VL30 derepression. iMEFs were transfected with a CRISPR-gRNA vector against *Setdb1* or *Trim28*. Two days later, transfected cells (tRFP-positive) were sorted using FACS Aria. Four days later, cells were harvested, and RT-qPCR of VL30 U3 class I was performed. (*Trim28*; n = 3 biological replicates *Setdb1*; n = 3 technical replicates) Error bars represent s.d. **0.0005 < P < 0.005, ***P < 0.0005, Student's *t*-test. Efficiencies of CRISPR KO were examined by immunoblotting with anti-Setdb1 or anti-Trim28 antibodies. **d** Reintroduced Setdb1 represses VL30 in *Setdb1* KO iMEFs. Setdb1 was stably expressed in *Setdb1* long-term-cultured KO iMEFs. RT-qPCR of VL30 U3 class I was performed. (n = 2 biological replicates). Error bars represent s.d. **0.0005 < P < 0.005, ***P < 0.0005, Student's *t*-test. Expression level of Setdb1 was examined by immunoblotting

## Discussion

In this study, we found that Setdb1 has a constant role in ERV silencing, which serves as a layer of epigenetic silencing not only in early embryonic or germ-lineage cells, but also in further differentiated somatic cells, even though a subset of cell-type-specific ERVs are only derepressed in Setdb1-deficient settings. Other transcriptional mechanisms, including cell-type-specific TFs, may restrict the expression of ERVs in Setdb1 KO cells (Fig. 6).

As described, distinct ERV families were derepressed in different cell types when Setdb1 was depleted (Fig. 1a). However, the enrichment of H3K9me3 on ERVs targeted and repressed by Setdb1 in mESCs was mostly maintained in various differentiated somatic cells. This situation was similar to that of VL30, whose expression was highly induced in iMEFs and GMPs, but not in ESCs, after Setdb1 depletion. These results suggest that Setdb1 continuously targets similar ERV classes and deposits H3K9me3 marks, regardless of the developmental stage or cell type and the expression level of Setdb1. It is then necessary to determine how different types of ERVs are activated in different cell types after Setdb1 depletion. Two distinct mechanisms are probably critical. According to one mechanism, ERVs are repressed by multiple epigenetic pathways or marks, and they may function redundantly with Setdb1 to silence some ERVs. Setdb1-mediated H3K9 methylation and Dnmt-mediated DNA methylation is an example of this mechanism. IAPEz is redundantly repressed by Setdb1 and DNA methylation in mESCs[4]. In addition to DNA and H3K9 methyltransferases, many other epigenetic or chromatin factors, such as Lsd1/Kdm1, the polycomb repressive complex 2, Yin yang 1 (Yy1), Erb3-binding protein 1 (Ebp1), and CAF-1, have roles in ERV silencing[42–46]. Therefore, it is possible that these factors and deposited epigenetic marks may exhibit crosstalk and interact to cause ERV silencing in a context-dependent manner. In the second mechanism, tissue-type-specific or cell-type-specific TFs restrict the transcriptional competence of each ERV. Previously, we found that MLV-type ERVs are specifically derepressed in B cells after Setdb1 depletion[19]. In this case, the expression of B-cell lineage-specific TF Pax5 confers competence for such an MLV

transcription. As shown in the current analyses, the cell-type-specific derepression of VL30 is also regulated by multiple cell-type-specific TF signaling pathways (Fig. 3). Thus, quiescent Setdb1-targeted ERVs have the potential for derepression upon inactivation of Setdb1, if their transcriptional machineries are competent.

Although it is generally recognized that DNA methylation, and not Setdb1-mediated H3K9me3, dominantly contributes to or is essential for ERV silencing in differentiated embryonic or adult somatic cells, only limited classes of ERVs, mostly IAPEz and its flanking LTR, IAPLTR1_Mm, are derepressed in Dnmt1 KD or KO iMEFs. Since IAPEz is stably and highly methylated, and this hyper DNA methylation is maintained in Setdb1 KO ESCs[3], it is reasonable to assume that IAPEz is not derepressed by Setdb1 KO alone in iMEFs. IAPEz is a high-copy-number ERV in the mouse genome (∼5000 copies, either full length or internally deleted), compared to members from other ERV families. Thus, the regulation of IAP was somewhat misleading with respect to the general behavior of ERVs.

In MEFs, only 1% (two genes) of gene promoters upregulated in Setdb1 KO iMEFs were marked by H3K9me3 (Supplementary Fig. 3a), indicating that only a minority of induced genes are directly controlled by Setdb1. This finding is consistent with the results obtained for ESCs, but far fewer genes are regulated by Setdb1 in MEFs[4]. The majority of upregulated genes in Setdb1 KO MEFs were IFN pathway-related genes. A recent work by Cuellar et al. showed that the loss of Setdb1 induces the derepression of retrotransposable elements, and the generated double-stranded RNAs activate the cytosolic RNA-sensing IFN signaling pathway[47]. The depletion of Dnmt1 using RNAi or 5-Aza-dC treatment also induced IFN pathway-related genes (Supplementary Fig. 10). Thus, these upregulated genes are most likely indirectly induced by the derepression of ERVs.

Other H3K9me3-specific methyltransferases are Suv39h1/2. Suv39hs are responsible for H3K9me3 deposition at the pericentric heterochromatin, containing major satellite repeats. They also deposit H3K9me3 at intergenic major satellite repeats, but only if intact consensus repeat sequences are maintained[24,48]. Previous genome-wide analyses in ESCs, NPCs, and iMEFs have not provided clear evidence of a major function of Suv39h-dependent H3K9me3 in directing gene transcription[24]. Thus, a main function of H3K9me3 marks is the repression of retro-elements, such as exogenous retroviruses, ERVs, and LINE elements, or imprinted genes, which are also thought to be derived from retrotransposons, but not coding genes. This is consistent with the role of H3K9 methylation in other species, such as fission yeast and plants[49,50].

ERVs are repressed by distinct, co-operative epigenetic mechanisms during the first few days of embryogenesis[15]. KRAB-ZFP family members are implicated in this process. KRAB-ZFPs bind to specific ERV loci and recruit their cofactors Trim28, Setdb1, and other silencing factors[15]. Zfp809, which belongs to the KRAB-ZFP family, represses VL30 by binding to PBS-pro, which is used by some retroviruses to prime reverse transcription[38]. Zfp809 is required to initiate ERV silencing during embryonic development, but becomes largely dispensable in differentiated somatic tissues. Zfp809 functions to repress only ERVs with the PBS-pro sequence. Thus, Zfp809 regulates small fractions of VL30 repressed by Setdb1 (Fig. 5a). For the silencing of ERVs without the PBS-pro sequence, other KRAB-ZFPs may bind to several loci of ERVs[51]. Trim28 KO MEFs or acute depletion of Trim28 in iMEFs leads to the marginal derepression of VL30 (Figs. 5b, c). Thus, the KRAB-ZFPs-Trim28 system is mostly dispensable for the maintenance of Setdb1-mediated VL30 silencing in MEFs. Lastly, it has been reported that a few number of VL30 elements were derepressed by treatment with a

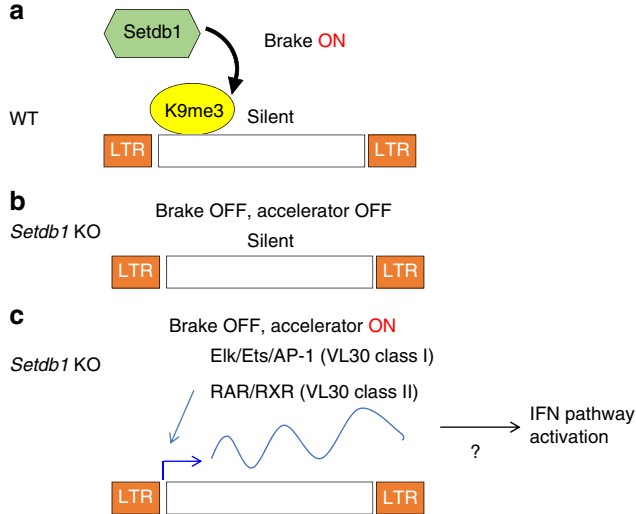

**Fig. 6** Model of Setdb1 function in ERV silencing. **a** Setdb1 deposits H3K9me3 at ERV loci generally in differentiated cells, not only in ESCs. **b** In Setdb1 KO cells H3K9me3 is reduced at ERV loci. However, TFs are required for each ERVs to get depressed. Without accelerator (TFs) ERVs are not expressed even without brake (H3K9me3). **c** For example, in Setdb1 KO iMEFs derepression of VL30 class I require Elk, Ets, and AP-1 TFs. VL30 class II is expressed only in the presence of RA in Setdb1 KO iMEFs. This ERV derepression might cause IFN pathway activation[47]

HDAC inhibitor, trichostatin A (TSA)[52]. We also performed an RNA-seq analysis of TSA-treated iMEFs, and could reproduce the reported findings. Interestingly, the impact of TSA was dominantly on the non-PBS-pro type elements (Supplementary Fig. 11). Thus, there must be more complex mechanisms for VL30 silencing. Future screen-based studies should be useful to identify genes or epigenetic factors responsible for Setdb1-mediated VL30 silencing in *Trim28* KO iMEFs.

## Methods

**Mouse embryonic fibroblasts.** *Setdb1* cKO iMEFs were described earlier[3]. Single cells were recloned by serial dilution. Setdb1 deletion was induced by 800 nM 4OHT treatment for 4 days and further cultured without 4OHT for 2 days. Setdb1 expression was determined by western blot analysis to screen clones that responded well (MEF3-12). Anti-Setdb1 antibody (Cell Applications Inc., CP10377; dilution: 1:1,000) was used. To generate long-term-cultured *Setdb1* KO iMEFs, 800 nM 4OHT was added to MEF3-12 for 4 days, and further cultured without 4OHT for more than 1 week. For the CRISPR-gRNA knockout, pL-CRISPR-EFS-tRFP plasmids[53] were transfected into iMEFs (MEF3-12) using the MEF2 Nucleofector Kit (Lonza). tRFP-positive cells were sorted using FACS Aria. The efficiency of knock out was confirmed by a western blot analysis with anti-Trim28 antibody (Abcam, ab22553; dilution: 1:1,000). Anti-α tubulin antibody (Sigma, T5168; dilution: 1:5,000) was used as an internal control. For chemical treatments, iMEFs were treated with 1 µM atRA (Sigma) for last 2 days, 1 µM MEK inhibitor PD0325901 (Wako) for last 24 hr, 5 µM Ezh2 inhibitor GSK126 (Xcess Bio) for last 5 days, and 150 nM TSA for 24 hr before cells were harvested. For Setdb1 rescue experiment in *Setdb1* KO iMEFs pPB-Setdb1-IRES Bsd plasmid with pPBase plasmid was transfected. Cells were selected with 8 µg ml⁻¹ Blastcidin (InvivoGen). To generate 3× Flag tagged Ets1 expressing *Setdb1* cKO iMEFs, pPB-3xF-Ets1-IRES puro plasmid with pPBase plasmid was transfected. Cells were selected with 1.5 µg ml⁻¹ puromycin (InvivoGen).

**Western blot analysis.** Cells were lysed in RIPA buffer (50 mM Tris-HCl (pH 7.5), 150 mM NaCl, 0.5 % NP-40, 0.5 % Na-deoxycholate, and 0.1 % SDS) supplemented with protease inhibitor cocktails (100×) (Nacalai) and 0.1 mM PMSF (Nacalai). Western blot analyses were performed using the antibodies as indicated. Uncropped scans of western blot images are available in Supplementary Fig. 12.

**RNA extraction and quantitative real-time RT-PCR.** RNA was isolated using RNeasy Plus Mini Kit (Qiagen). DNase I (NEB) was added to the column. Total RNA (1 µg) was used to generate cDNA, using the Omniscript RT Kit (Qiagen) and random primers. qPCR was carried out using Power SYBR Green PCR Master Mix (ABI) on the ABI StepOnePlus. The signals were normalized against Hprt signals. Primer sequences are shown in Supplementary Table 1.

**RNA-seq.** Approximately 54 to 73 million high-quality 75-bp paired-end reads per sample for the forebrain[18], ~110 million 100-bp paired-end reads per sample for iMEFs (Ctrl, 4OHT 5d and long-term cultured), 51-bp single-read per sample for iMEFs (siCtrl, siDnmt1, 4OHT 6d and 4OHT siDnmt1), and ~15 to 22 million 60-bp single-read for GMPs were mapped to the mouse genome (mm9) using the TopHat2 splice junction mapper (version 2.0.12)[54] with parameter (-g 1) telling the software to report best alignment only once for multi-hit reads. Mm9 Refseq gtf file was downloaded from UCSC (http://genome.ucsc.edu) in Oct 22, 2014 and applied as gene annotation file. After obtaining the aligned bam files, the Cufflinks algorithm (version 2.2.1)[55] was used to calculate FPKM with the Refseq gtf file. Repeat elements were downloaded from the University of California at Santa Cruz RepeatMasker track (mm9) and we selected ERVs annotated with high SW (Smith–Waterman) probability scores (≥2000). Sequences of RepeatMasker annotated ERVs were compared with Repbase consensus sequences and scored for percentage of similarity or length mismatch. We applied SW 2000 as a cutoff because sufficient LTR only types (such as IAPLTR1a, RLTR6_Mm, or RLTR10C) could not be collected with high SW (>3000). Differential expression was computed using BEDTools for ERVs (http://bedtools.googlecode.com) with a minimum of 1-bp overlap. According to published data[28], MMVL30 subclasses were assigned to their neighboring RLTR6 repeats and reconfirmed the position from which each LTR starts by alignment with consensus sequence. The genomic positions of ERVs used for the analysis are shown in Supplementary Data 4.

**Native ChIP and crosslinked ChIP.** Native ChIP assays were performed as described previously[3]. A mouse monoclonal antibody against H3K9me3 (2F3) was used[56].

The antibody was incubated with anti-mouse IgG Dynabeads (Veritas for 1 h on ice and overnight after the addition of MNase-digested chromatin. For crosslinked ChIP, 1 × 10⁷ cells were crosslinked with 1% formaldehyde at 25 °C for 10 min. Chromatin was extracted and then sonicated using Bioruptor USD-250 to obtain an average fragment size of 300–500 bp. Immunoprecipitation was performed

using Dynabeads with antibodies, followed by purification using QIAquick PCR Purification Kit (Qiagen).

**Bisulfite sequencing.** Genomic DNA was purified from MEFs and bisulfite-converted using MethylCode Bisulfite Conversion Kit (Thermo). PCR products were amplified using TaKaRa EpiTaq HS (TaKaRa) and subcloned using TOPO Cloning Kit (Thermo) for sequencing. CpG methylation was analyzed using the QUMA tool (http://quma.cdb.riken.jp).

**siRNA.** For knockdown experiments, 50 nM siRNAs targeting *Dnmt1* (siGENOME SMARTpool) or control siRNA (Dharmacon) was transfected into *Setdb1* cKO iMEFs using RNAiMAX (Thermo). Transfected cells were passaged 24 h after transfection. A second transfection was conducted with the same reagents on the following day. Cells were harvested at day 6 and RNA was isolated. For simultaneous KO of Setdb1, 4OHT was added at day 0.

**ChIP-seq.** For ChIP-seq of H3K9me3, a polyclonal antibody against H3K9me3 (abcam ab8898) was used. For library preparation, KAPA Hyper Prep kit (KAPA biosystems) was used. Approximately 200 million 50-bp paired-end reads for forebrain cells, and ~20 to 23 million 76-bp single-end reads for iMEF were mapped to the mouse genome (mm9) using the Bowtie2 short read aligner (version 2.2.3) with default parameters[57]. A multi-hit read was assigned to one site randomly selected from among valid alignments and duplicate reads were removed using Picard tools (https://broadinstitute.github.io/picard/).

**Public sequencing data.** Raw reads were downloaded from publicly available ChIP-seq (H3K9me3 in ESC: GSM1375155, GSM727425), (H3K9me3 in MEF: GSM1375168, GSM1375173), (H3K9me3 in GMP: DRX021712, DRX021713, DRX021716, DRX021717), and RNA-seq (*Setdb1* KO ESC established from conditional KO ESC: GSM727423, GSM727424), (*Dnmt1* cKO ESC and *Dnmt1, Setdb1* double cKO ESC: GSE77781), (*Dnmt1* WT, *p53-/-* and *Dnmt1-/-*, *p53-/-* iMEF: GSM1089793, GSM1089794), (*Zfp809* KO MEF established from KO embryo at E12.5: SRX487521), (*Trim28* KO MEF established from *Trim28* conditional KO MEF: GSM1916177, GSM1916178, GSM1916179, GSM1916180). Mapping and data processing were performed as described above.

**ChIP-seq NGS plots.** NGS plots show the average ratios of normalized read density between ChIP and input samples every 10 bp (1 bin) of H3K9me3 from −5 kb to 10 kb around genomic ERV elements in ESC, forebrain, iMEF, and GMP. We selected ERVs containing -int element (which means the internal region) with flanked LTRs. Position 1 is 5′ start site of the -int element and average length of flanking LTR elements is about 600 bp. Positions of LTRs and int for each ERV element are indicated below the plots. Unique ERVs containing MMERVK10C-int (167 copies, average length: 7190 bp, flanking LTRs are RLTR10C), IAPEz-int (593 copies, average length: 5516 bp, flanking LTRs are IAPLTR1a), RLTR4_Mm-int (50 copies, average length: 7230 bp, flanking LTRs are RLTR4_MM), RLTR6-int (186 copies, average length: 7212 bp, flanking LTRs are RLTR6_Mm), and MMVL30-int (86 copies, average length: 5294 bp, flanking LTRs are RLTR6_Mm) were analyzed. ChIP-seq data sets for ESC and iMEF[24], GMP[20] and forebrain cells (this study) were used. Alignment files for H3K9me3 ChIP-seq data (bam format) were transformed to read coverage files (bigwig format) and processed to obtain plots for each ERV of interest using DeepTools2.0[58] with the following parameters: -ratio log2, bin Size 10, scaleFactors SES. Read densities are normalized with read depth and million mapped reads of each NGS data.

**Data availability.** All NGS sequencing data that support the findings of this study are available in the NCBI Gene Expression Omnibus (https://www.ncbi.nlm.nih.gov/geo/) database under the accession number GSE102490.

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

## Acknowledgements

We thank Ms. Chikako Shimura for her technical assistance with the western blot analysis and all other members of the Shinkai lab for their experimental support, critical feedback, and suggestions. We also thank Mr. Kenji Ohtawa (RIKEN BSI Research Resources Center: RRC) who provided technical support for cell sorting and Drs. Eugene M. Oltz, Patrick L. Collins, and Matthew C. Lorincz for critically reading the manuscript. Illumina sequencing was supported by Genome Resource and Analysis Unit RIKEN CDB. This work was supported in part by AMED-CREST and a RIKEN internal research fund.

## Author contributions

M.K. designed and conducted the experiments, K.T. designed the experiments and performed informatics analysis, and Y.S. designed and supervised the experiment. M.K., K.T., and Y.S. wrote the manuscript.

## Additional information

**Competing interests:** The authors declare no competing interests.

