## [Peer Review File · Nature Communications]

Reviewers' comments:

Reviewer #1 (Remarks to the Author):

Kato, Takemoto, and Shinkai's article details a compelling dissection of the roles of histone and DNA methylation in the silencing of endogenous retroviruses. Through conditional and CRISPR-based knock-outs, the researchers identify Setdb1-mediated repression of retroelements such as IAPez and MMVL30-int in differentiated cells. They also elucidate distinct upstream mechanisms repressing elements containing and lacking PBS-pro, noting that while Setdb1 is necessary for controlling both subsets of viral elements, Zfp809 is only involved in silencing of PBS-pro containing sequences. The paper also strengthens the argument for Dmmt1's involvement in long-term suppression of ERVs, and proposes a role for Setdb1 in the interferon response. Although the paper makes strong arguments for most of its claims a few of the claims should be strengthened.

1. The finding that Setdb1 depletion in MEFs leads to a potential activation of the innate immunity pathway is an interesting result, particularly since retrotransposon activation has been suggested to be a contributing factor to Aicardi Goutieres Syndrome and immune system activation by DNA methylation inhibitors as a therapeutic intervention in cancer. But the mechanism is not explored. Although it is plausible as is suggested that this activation is due to increased expression of retrotransposons, this idea is not well developed. Zfp809 KO MEFs for example also display reactivated VL30 elements, but there is no activation of innate immunity genes. Is it possible that DNA damage associated with SETDB1 loss that is stimulating the innate immune pathway? Or is it one particular family of retrotransposon? Is it reversible with RT inhibitors? Can RNA/DNA hybrids be detected in the cytoplasm (using S9.6 antibody). Is it cGas/STING dependent? This should be more developed if it is to be placed in the model figure. Otherwise it is entirely too speculative.

2. I am not sure the long-term culture experiment adds much to the story. The authors show that there is attenuation of the activation of VL30 after long term culture, but their subsequent experiments do not add new insights into the mechanism (they only show that neither DNA methylation nor H3K27me3 are responsible.) Thus I feel this should either be further explored or should be removed from the manuscript.

Minor Comments

1. I have mixed feelings about the use of the word "deplete" to describe the CRISPR knock-outs as somewhat confusing, as I tend to associate depletion with siRNA or degran-mediated protein loss. Cells are either KO or not, and it is clear that it is a mixed population of WT and KO since there is still detectable protein.

2. Typo: "te" in place of "the" page 21 line 362

3. Typo: "PRKM" in place of "RPKM" page 44 line 773

Reviewer #2 (Remarks to the Author):

Kato, Takemoto and Shinkai investigated the role of SETDB1-mediated silencing of transposable elements (TEs) in MEFs. They found that, although SETDB1 is important for H3K9me3 maintenance at many TE classes, only a subset of these are derepressed upon SETDB1 depletion. In the case of VL30 elements, this is explained by differences in transcription factor requirements. They also found that the ZFP/KAP1 pathway is not necessary to maintain TE repression in MEFs.

This is a clear, well put together study that brings in some interesting insights into the role of SETDB1 in TE regulation in differentiated cells. Personally, I think the most interesting data relate

to transcription factor binding sites at derepressed VL30 elements. The experiments on DNMT1, ZFP809 and KAP1 complement the remaining manuscript well.

I have a few specific comments that I believe would improve the manuscript, namely by adding further support to the role of transcription factors at VL30 elements:

1. In the experiment using a MAPK inhibitor (Fig. 4e), VL30 elements with mutated Elk binding sites are expected to be equally affected by SETDB1 depletion with or without MAPK inhibition. Could the authors test this? RT-qPCR primers could be designed for specific VL30 copies or, alternatively, primers that bind to the Elk binding site could be used to differentiate between the two pools.
2. A ChIP for Elk/Ets1 would strengthen the authors' hypothesis. Again, using specific primers, it should be possible to detect binding of the relevant transcription factors at the derepressed VL30 copies. The MAPK inhibition experiment would also be useful here to demonstrate that the ChIP is specific and that Elk binding is lost upon MAPK inhibition.
3. Several key analyses are performed using pipelines that include multi-hit reads. This is fine, but it should be made clear in the main text. More importantly, plotting heatmaps with this type of alignment is misleading (Fig. 2c-d), as it gives the false impression that all TE copies have similar H3K9me3 profiles, when in reality what is plotted is by and large a random assignment of reads to different copies. It would be preferable to use only the overall trend plots (which the authors call 'NGS' plots – not sure what this stands for).
4. The authors report that half of the upregulated genes have ERVs nearby, which gives the impression that there is some meaningful association, whereas their data largely suggest that this is not related to SETDB1 regulation. Is the percentage of ERVs seen at these genes simply what would be expected by chance? Do down regulated genes or unchanged genes display similar percentages of ERVs?

Reviewer #3 (Remarks to the Author):

The histone methyltransferase SETDB1, which deposits the repressive mark H3K9me3, was first shown to suppress families of endogenous retroviruses (ERVs) in mouse ES cells and it has subsequently been shown that SETDB1 also plays a role in ERV repression in at least some somatic cells such as B cells and neural progenitor cells. In this study, Kato et al investigate the role of SETDB1 in ERV silencing in mouse embryonic fibroblasts (MEFs) and finds that its reduction leads to transcriptional activation of some ERV families, particularly VL30 elements, which is dependent on cell type specific transcription factors. While not particularly novel or unexpected, the study is generally well done and comprehensive and makes a substantial contribution to current literature on epigenetic mechanisms controlling ERV transcription. The following points should be addressed by the authors:

1. The authors have not adequately discussed the prior relevant literature on VL30 elements and integrated these previous findings into their study. In particular, a key, relevant paper is not mentioned: PLoS Genet. 2010 Apr 29;6(4):e1000927. "Epigenetic regulation of a murine retrotransposon by a dual histone modification mark." by Brunmeir R et al. This paper reports upregulation of VL30 upon HDAC inhibition and states in their summary:

"We found that one LTR retrotransposon family encompassing virus-like 30S elements (VL30) showed significant histone H3 hyperacetylation and strong transcriptional activation in response to TSA treatment. Analysis of VL30 transcripts revealed that increased VL30 transcription is due to

enhanced expression of a limited number of genomic elements, with one locus being particularly responsive to HDAC inhibition. Importantly, transcriptional induction of VL30 was entirely dependent on the activation of MAP kinase pathways, resulting in serine 10 phosphorylation at histone H3.”

This is very relevant since SETDB1 associates with HDACs. How do the present findings relate to the Brunmeir results? Also, Brunmier found that only very few VL30 elements were greatly upregulated with TSA treatment. Kato et al should determine if they observe the same elements upregulated upon SETDB1 depletion, which may help explain mechanisms.

2. Another paper on VL30 elements should also be mentioned: Mob DNA. 2016 May 6;7:10. doi: 10.1186/s13100-016-0066-8. eCollection 2016. Genomic analysis of mouse VL30 retrotransposons. By Markopoulos G et al. This paper also gives a good overview of what is known about VL30 elements in their Introduction. For example, it cites several papers that discuss the tissue-specificity of expression of different VL30 subtypes, based on TF binding site differences, a fact that has been known for a long time. Therefore the finding of Kato et al that derepression of VL30s is dependent on cell type specific transcription factors is of course expected and not particularly novel.

3. In the section on middle of page 7, the authors state: “In MEFs, ~50% of the H3K9me3-marked up-regulated genes (11 out of 27 genes) had ERV or LINE insertions with H3K9me3 enrichment. ” Is this statistically significant? What points are the authors trying to make? Is this more than one would expect by chance?

4. The DNA methylation analysis is confusing and incomplete. Figure 4C shows bisulfite sequencing of one specific IAP copy (in the Mnd1 gene). Why was this copy chosen? More importantly, it is unclear what the VL30 bisulfite represents but I believe this is just random clones/alleles from any VL30 copy. The same appears to be true for Supplementary Fig 6 but this is not made clear. To support the authors’ conclusion that there is no reduction in DNA methylation upon depletion of SETDB1, a few individual VL30 copies that do and do not become transcriptionally activated upon SETDB1 depletion should be measured for DNA methylation. Perhaps the few copies that are transcriptionally activated do indeed show reductions in DNA methylation.

5. Supplementary Table 4 needs the full references for the primer sequences.

Response to Reviewers' comments:

Reviewers' comments:

Reviewer #1 (Remarks to the Author):

Kato, Takemoto, and Shinkai's article details a compelling dissection of the roles of histone and DNA methylation in the silencing of endogenous retroviruses. Through conditional and CRISPR-based knock-outs, the researchers identify Setdb1-mediated repression of retroelements such as IAPEz and MMVL30-int in differentiated cells. They also elucidate distinct upstream mechanisms repressing elements containing and lacking PBS-pro, noting that while Setdb1 is necessary for controlling both subsets of viral elements, Zfp809 is only involved in silencing of PBS-pro containing sequences. The paper also strengthens the argument for Dmnt1's involvement in long-term suppression of ERVs, and proposes a role for Setdb1 in the interferon response. Although the paper makes strong arguments for most of its claims a few of the claims should be strengthened.

1. The finding that Setdb1 depletion in MEFs leads to a potential activation of the innate immunity pathway is an interesting result, particularly since retrotransposon activation has been suggested to be a contributing factor to Aicardi Goutieres Syndrome and immune system activation by DNA methylation inhibitors as a therapeutic intervention in cancer. But the mechanism is not explored. Although it is plausible as is suggested that this activation is due to increased expression of retrotransposons, this idea is not well developed. Zfp809 KO MEFs for example also display reactivated VL30 elements, but there is no activation of innate immunity genes. Is it possible that DNA damage associated with SETDB1 loss that is stimulating the innate immune pathway? Or is it one particular family of retrotransposon? Is it reversible with RT inhibitors? Can RNA/DNA hybrids be detected in the cytoplasm (using S9.6 antibody). Is it cGas/STING dependent? This should be more developed if it is to be placed in the model figure. Otherwise it is entirely too speculative.

Response: The reviewer #1's comment is reasonable that we should provide more mechanistic insight into the *Setdb1* KO-mediated IFN pathway activation if we propose the model as shown in the original Fig. 7. However, unfortunately or fortunately (?), the mechanistic issue of the Setdb1-mediated IFN signaling pathway regulation was significantly clarified by Cuellar et al. (J Cell Biol. 2017 Nov 6;216(11):3535-3549.) after we submitted our manuscript to Nat Commun. In this paper, they showed that loss of *SETDB1* in AML cells triggers desilencing of retrotransposable elements that leads to the production of double-stranded RNAs (dsRNAs) and this is coincident with induction of a type I interferon response through the dsRNA-sensing pathway. Therefore, we cited this work in the revised manuscript and legend of Figure 7

to make clear that “→IFN pathway activation” is based on this work, but did not perform additional studies on this issue by our hands. Hopefully, this description is acceptable.

Although Cuellar et al. work is nice, but we still could not explain some of the findings of the IFN pathway activation such as 1) we did not see IFN pathway activation in *Zfp809* KO MEFs as pointed out by the reviewer, 2) We saw the IFN pathway activation in *Setdb1* and *Dnmt1* KD iMEFs but not in TSA treated iMEFs although derepression of VL30 is similar to that in *Setdb1* KO iMEFs (Supplementary Fig. 9), 3) as the reviewer suggested, it is possible that DNA damage response is involved in the *Setdb1* loss phenotypes because we observed gamma H2A foci in *Setdb1* KO iMEFs (not shown in the manuscript). We hope to clarify these unsolved problems in the future studies.

2. I am not sure the long-term culture experiment adds much to the story. The authors show that there is attenuation of the activation of VL30 after long term culture, but their subsequent experiments do not add new insights into the mechanism (they only show that neither DNA methylation nor H3K27me3 are responsible.) Thus I feel this should either be further explored or should be removed from the manuscript.

Response: We agree with the reviewer’s comment that we could not figure out why the activation of VL30 goes down after long-term culture. However, these data are important for our manuscript to show that there is no clear contribution of DNA methylation in long-term cultured *Setdb1* KO iMEFs for VL30 transcriptional regulation. Therefore, we keep it in the revised manuscript but moved from main Figure to Supplemental data (revised Supplementary Fig. 8) and combined with EZH2 inhibitor data (Supplementary Fig. 7 in the original manuscript).

Minor Comments

1. I have mixed feelings about the use of the word “deplete” to describe the CRISPR knock-outs as somewhat confusing, as I tend to associate depletion with siRNA or degran-mediated protein loss. Cells are either KO or not, and it is clear that it is a mixed population of WT and KO since there is still detectable protein.

Response: We deleted the word “deplete or depletion” for description of the CRISPR knock-outs.

2. Typo: “te” in place of “the” page 21 line 362

Response: Thank you for pointing out the typo. We corrected it.

3. Typo: "PRKM" in place of "RPKM" page 44 line 773

Response: Thank you for pointing out the typo. We corrected it.

Reviewer #2 (Remarks to the Author):

Kato, Takemoto and Shinkai investigated the role of SETDB1-mediated silencing of transposable elements (TEs) in MEFs. They found that, although SETDB1 is important for H3K9me3 maintenance at many TE classes, only a subset of these are derepressed upon SETDB1 depletion. In the case of VL30 elements, this is explained by differences in transcription factor requirements. They also found that the ZFP/KAP1 pathway is not necessary to maintain TE repression in MEFs.

This is a clear, well put together study that brings in some interesting insights into the role of SETDB1 in TE regulation in differentiated cells. Personally, I think the most interesting data relate to transcription factor binding sites at derepressed VL30 elements. The experiments on DNMT1, ZFP809 and KAP1 complement the remaining manuscript well.

I have a few specific comments that I believe would improve the manuscript, namely by adding further support to the role of transcription factors at VL30 elements:

1. In the experiment using a MAPK inhibitor (Fig. 4e), VL30 elements with mutated Elk binding sites are expected to be equally affected by SETDB1 depletion with or without MAPK inhibition. Could the authors test this? RT-qPCR primers could be designed for specific VL30 copies or, alternatively, primers that bind to the Elk binding site could be used to differentiate between the two pools.◦

Response: First of all, thank you so much for the reviewer #2's positive evaluation on our manuscript and valuable comments.

We performed RNA-seq analysis for the MEK inhibitor (PD0325901) experiment as shown in original Fig. 3e. However, in contrast to the prediction, derepression of those VL30 elements with mutated Elk/Ets binding site(s) were also diminished by the MEK inhibitor treatment as similar to those containing intact Elk/Ets binding sites. Since AP-1 is also activated by the MAPK pathway and AP-1 binding site is mostly intact in the derepressed VL30 elements regardless of Elk/Ets binding site mutation(s) as shown in Fig. 3d and Supplemental Table 3. Thus, we speculate that AP-1 may also contribute to activation of VL30 U3 I elements, therefore derepression of those elements with mutated Elk/Ets binding sites were also affected by MEK inhibitor. We briefly stated these results in the revised manuscript. Just as *reference for the reviewers, we provided the MEK inhibitor RNA-seq analysis data as Supplemental material (Fig. SS1).*

“As previously reported, AP-1 is also present downstream of the MAPK pathway to activate VL30 elements ^{32,33}. Inhibition of MEK activity may also inhibit AP-1 activity.”
p13, line 14-15.

2. A ChIP for Elk/Ets1 would strengthen the authors' hypothesis. Again, using specific primers, it should be possible to detect binding of the relevant transcription factors at the derepressed VL30 copies. The MAPK inhibition experiment would also be useful here to demonstrate that the ChIP is specific and that Elk binding is lost upon MAPK inhibition.

Response: Unfortunately, we are no longer able to access to the Santa Cruz (sc-350) anti-Ets1 antibody which worked well for the ChIP experiment. Therefore, instead examining endogenous Ets1, we looked at exogenously expressed FLAG-tagged Ets1 by using *Setdb1* cKO iMEFs stably expressing FLAG-tagged Ets1 and performed ChIP-qPCR experiments. We observed enrichments of FLAG-Ets1 after *Setdb1* depletion at #9 loci (Ets1 site is intact) but not at #6 and #18 (Ets1 site is mutated) although the enrichment is not so strong. We included those data in supplemental Fig. 4b and stated the result in the revised manuscript.

“To test the direct involvement of Ets1 in VL30 derepression, we performed ChIP-qPCR analysis of Flag-tagged Ets1 to see an enrichment of Ets1 at VL30 loci. We observed a slight enrichment of Ets1 at U3 class I #9 loci (Ets binding site is intact) after *Setdb1* depletion, but no enrichment at #6 and #18 copies, in which the Ets-binding site is mutated (Supplementary Fig. 5), suggesting that Ets1 might contribute to VL30 derepression. However, the Ets family is large, and we do not rule out the possibility that other Ets proteins are involved in VL30 derepression.” P13, line15-P14, line3.

3. Several key analyses are performed using pipelines that include multi-hit reads. This is fine, but it should be made clear in the main text. More importantly, plotting heatmaps with this type of alignment is misleading (Fig. 2c-d), as it gives the false impression that all TE copies have similar H3K9me3 profiles, when in reality what is plotted is by and large a random assignment of reads to different copies. It would be preferable to use only the overall trend plots (which the authors call 'NGS' plots – not sure what this stands for).

Response: We totally agree with the reviewer #2's comments. We made clear that used pipelines for TE informatics analysis include multi-hit reads and also removed heatmaps from this figure.

“A multi-hit read was assigned to one site randomly selected from among valid alignments and duplicate reads were removed using Picard tools (<https://broadinstitute.github.io/picard/>).” P29, line 4-6.

4. The authors report that half of the upregulated genes have ERVs nearby, which gives the impression that there is some meaningful association, whereas their data largely suggest that this is not related to SETDB1 regulation. Is the percentage of ERVs seen at these genes simply what would be expected by chance? Do down regulated genes or unchanged genes display similar percentages of ERVs?

Response: We calculated the ratio between up-regulated genes with ERV insertion and without insertion and compared it with those for down-regulated and unchanged genes. Our conclusion is that there was no statistically significant difference between up-regulated genes and others. To avoid giving the impression that there is some meaningful association, we corrected the text.

“Furthermore, we calculated the ratio between upregulated genes with and without ERV insertions, and compared it with those for the downregulated and unchanged genes. There was no statistically significant difference between the values for the upregulated genes and those for the other genes (The ratio of up-regulated genes is 0.0472, down-regulated 144 genes 0.0434 and unchanged 22533 genes 0.0467, thus ERVs seen at up-regulated genes were expected by chance ($p = 0.12$)).” P7, line 12-18.

Reviewer #3 (Remarks to the Author):

The histone methyltransferase SETDB1, which deposits the repressive mark H3K9me3, was first shown to suppress families of endogenous retroviruses (ERVs) in mouse ES cells and it has subsequently been shown that SETDB1 also plays a role in ERV repression in at least some somatic cells such as B cells and neural progenitor cells. In this study, Kato et al investigate the role of SETDB1 in ERV silencing in mouse embryonic fibroblasts (MEFs) and finds that its reduction leads to transcriptional activation of some ERV families, particularly VL30 elements, which is dependent on cell type specific transcription factors. While not particularly novel or unexpected, the study is generally well done and comprehensive and makes a substantial contribution to current literature on epigenetic mechanisms controlling ERV transcription. The following points should be addressed by the authors:

1. The authors have not adequately discussed the prior relevant literature on VL30 elements and integrated these previous findings into their study. In particular, a key, relevant paper is not mentioned: PLoS Genet. 2010 Apr 29;6(4):e1000927. "Epigenetic regulation of a murine retrotransposon by a dual histone modification mark." by Brunmeir R et al. This paper reports upregulation of VL30 upon HDAC inhibition and states in their summary:

"We found that one LTR retrotransposon family encompassing virus-like 30S elements (VL30) showed significant histone H3 hyperacetylation and strong transcriptional activation in response to TSA treatment. Analysis of VL30 transcripts revealed that increased VL30 transcription is due to enhanced expression of a limited number of genomic elements, with one locus being particularly responsive to HDAC inhibition. Importantly, transcriptional induction of VL30 was entirely dependent on the activation of MAP kinase pathways, resulting in serine 10 phosphorylation at histone H3."

This is very relevant since SETDB1 associates with HDACs. How do the present findings relate to the Brunmeir results? Also, Brunmeir found that only very few VL30 elements were greatly upregulated with TSA treatment. Kato et al should determine if they observe the same elements upregulated upon SETDB1 depletion, which may help explain mechanisms.

We performed RNA-seq analysis of TSA treated iMEFs to see which copies of VL30 U3 I are derepressed. We observed several copies of VL30 U3 I were derepressed including copies which Brunmeir et al have reported (NT_039207 (#9), NT_039649 (#63) and NT_039341 (#21)). However, we observed very weak derepression of PBS-pro copies in TSA treated iMEFs. Similarly, a slight derepression of VL30 U3 I copies by *Dnmt1* KD was observed, but VL30 PBS-pro copies were not included (Fig. 4c). Those results suggest that silencing of PBS-pro copies is dependent on *Setdb1*- but not *Dnmt1*- or HDACs-mediated pathway. We stated these results in the text including

citation of Brunmeir R et al paper and included the TSA and Dnmt1 data as revised Supplementary Fig. 9 and Fig. 4c.

“Lastly, it has been reported that a few number of VL30 elements were derepressed by treatment with a HDAC inhibitor, trichostatin A (TSA) ⁵². We also performed an RNA-seq analysis of TSA-treated iMEFs, and could reproduce the reported findings. Interestingly, the impact of TSA was dominantly on the non-PBS-pro type elements (Supplementary Fig. 9).” P24, line 5-9.

2. Another paper on VL30 elements should also be mentioned: Mob DNA. 2016 May 6;7:10. doi: 10.1186/s13100-016-0066-8. eCollection 2016. Genomic analysis of mouse VL30 retrotransposons. By Markopoulos G et al. This paper also gives a good overview of what is known about VL30 elements in their Introduction. For example, it cites several papers that discuss the tissue-specificity of expression of different VL30 subtypes, based on TF binding site differences, a fact that has been known for a long time. Therefore the finding of Kato et al that derepression of VL30s is dependent on cell type specific transcription factors is of course expected and not particularly novel.

Response: Following to the comment, we cited Markopoulos G et al. paper in the revised manuscript. Thank you!

“The structural analysis of LTRs of VL30 has been reported to show a possible requirement of tissue-specific TFs ²⁹.” P13, line 1-2.

3. In the section on middle of page 7, the authors state: “In MEFs, ~50% of the H3K9me3-marked up-regulated genes (11 out of 27 genes) had ERV or LINE insertions with H3K9me3 enrichment. “ Is this statistically significant? What points are the authors trying to make? Is this more than one would expect by chance?

Response: Same comment is provided by the reviewer #2. We re-calculated the ratio between up-regulated genes with ERV insertion and without insertion and compared it with those for down-regulated and unchanged genes. Our conclusion is that there was no statistically significant difference between up-regulated genes and others. To avoid giving the impression that there is some meaningful association, we corrected the text.

“Furthermore, we calculated the ratio between upregulated genes with and without ERV insertions, and compared it with those for the downregulated and unchanged genes. There was no statistically significant difference between the values for the upregulated genes and those for the other genes (The ratio of up-regulated genes is 0.0472, down-regulated 144 genes 0.0434 and unchanged 22533 genes 0.0467, thus ERVs seen at up-regulated genes were expected by chance ($p = 0.12$)).” P7, line 12-18.

4. The DNA methylation analysis is confusing and incomplete. Figure 4C shows bisulfite sequencing of one specific IAP copy (in the *Mnd1* gene). Why was this copy chosen? More importantly, it is unclear what the VL30 bisulfite represents but I believe this is just random clones/alleles from any VL30 copy. The same appears to be true for Supplementary Fig 6 but this is not made clear. To support the authors' conclusion that there is no reduction in DNA methylation upon depletion of SETDB1, a few individual VL30 copies that do and do not become transcriptionally activated upon SETDB1 depletion should be measured for DNA methylation. Perhaps the few copies that are transcriptionally activated do indeed show reductions in DNA methylation.

Response: Reviewer #3 pointed out very important issue. The reason we chose *Mnd1* locus is we studied this locus very intensively. IAPEz from this locus is expressed in *Setdb1* KO ESCs and forebrain in addition to in *Dnmt1* KD iMEFs. As the reviewer pointed out, it is better to look at DNA methylation status of individual loci of VL30 in *Setdb1* KO iMEFs. Thus, first we decided to present the expression of individual loci of VL30 U3 I in *Dnmt1* KD iMEFs. #38 is derepressed in *Dnmt1* KD strongly and #21 is derepressed mildly. Then we performed bisulfite analysis of those copies in *Setdb1* KO iMEFs. Even though #38 and #21 are transcriptionally activated in *Setdb1* KO iMEFs, DNA methylation status are not changed. Thus, we conclude that there is no clear reduction of DNA methylation upon depletion of *Setdb1*, at least at these specific VL30 loci. We described this in the result section and the result is included as Supp. Fig. 5b.

"*Dnmt1* siRNA treatment significantly reduced CG methylation levels on IAPEz (at the *Mnd1* locus) and VL30 (PBS-pro) (Supplementary Fig. 7a); however, we did not detect clear reduction in DNA methylation at the VL30 loci (individual locus of #21, #38 and pool of PBS-pro elements loci) after the depletion of *Setdb1* (Supplementary Fig. 7b)."
P16, line 6-9.

5. Supplementary Table 4 needs the full references for the primer sequences.

Response: we cited references for the primer sequences we used.

Reviewers' comments:

Reviewer #1 (Remarks to the Author):

This is a nice paper that extends our understanding of proviral silencing in somatic cells by Setdb1. It also adds to our understanding of the necessity of TFs to activate LTRs in the absence of silencing factors for activation of RTEs. I recommend publication in its current form.

Reviewer #2 (Remarks to the Author):

The authors have done additional experiments and made adjustments to the manuscript in a way that largely answer my queries in a satisfactory manner and improve the manuscript. I just have two small comments/suggestions on the new experiments:

1) I would encourage the authors to include the MAPK inhibition RNA-seq data in the manuscript and to publicly release it.

2) The Ets1 ChIP (Supp. Fig. 5) was performed on 3 "technical experiments". Can the authors please clarify the meaning of this? Were they independent biological samples? Independent ChIP experiments on the same biological material? Or do the error bars simply refer to the qPCR variability from a single ChIP? The answer to this has implications on how reliable the data are and therefore how much weight it should have in supporting the mechanistic model.

Reviewer #3 (Remarks to the Author):

In the revision the authors have answered my queries and concerns in a satisfactory manner except for their response to my original point 4 about DNA methylation. As requested, they performed bisulfite DNA methylation analysis on two individual VL30 copies (copy 21 and copy 38) before and after SETDB1 knockdown but, curiously, they do not show all the CpGs for these copies in Supplementary figure 7b. These copies contain many more CpG sites and all should be assayed. It appears from the CpG spacing in the figure that the sites shown are all upstream of the probable TATAA box, whereas, at least for one of the copies, there are 8 additional CpGs downstream of the TATAA box. In this reviewer's experience, methylation status of CpGs downstream of the TATAA box is most likely to correlate with transcriptional activity of LTRs. The LTRs are not so long so it should be not be a major technical challenge to amplify the whole LTR by placing one primer in the 5' interior of the VL30 and one in the unique flanking region. As it stands, the authors' conclusion "that there is no clear reduction of DNA methylation upon depletion of Setdb1, at least at these specific VL30 loci" cannot be made. This is an important point.

Response to Reviewers' comments:

Reviewer #2 (Remarks to the Author):

The authors have done additional experiments and made adjustments to the manuscript in a way that largely answer my queries in a satisfactory manner and improve the manuscript. I just have two small comments/suggestions on the new experiments:

- 1) I would encourage the authors to include the MAPK inhibition RNA-seq data in the manuscript and to publicly release it.

Response: Thank you for the reviewer's encouragement to include the MEK inhibitor treatment RNA-seq data in our manuscript. Now, we included those data as Supplementary Fig 5a, stated the result and deposited it to GEO.

"RNA-seq analysis showed that induction of low derepressed VL30 elements with mutated Elk/Ets binding sites such as #47 and #58 copies were also diminished by the MEK inhibitor treatment as similar to those containing intact Elk/Ets binding sites (Supplementary Fig. 5a). Indeed, AP-1 is known to be present downstream of the MAPK pathway to activate VL30 elements^{32, 33} and AP-1 binding site is mostly intact in the derepressed VL30 elements regardless of Elk/Ets binding site mutation(s) (Fig. 3d and Supplemental Table 3). Thus, we speculate that AP-1 also contribute to activation of VL30 U3 class I elements after Setdb1 depletion." P14, line 2-3.

- 2) The Ets1 ChIP (Supp. Fig. 5) was performed on 3 "technical experiments". Can the authors please clarify the meaning of this? Were they independent biological samples? Independent ChIP experiments on the same biological material? Or do the error bars simply refer to the qPCR variability from a single ChIP? The answer to this has implications on how reliable the data are and therefore how much weight it should have in supporting the mechanistic model.

Response: In this Ets1 ChIP-qPCR data, the error bars simply refer to the qPCR variability from a single ChIP experiment. However, we repeated this experiment 3 times using independent biological samples and results were reproducible. We stated this in the supplementary fig

legend. For the reviewer's reference, we provided all of those Flag-tagged Ets1 ChIP-qPCR data (Kato et al, Fig. SS2). Because the values of % input were varied among experiments, we could not calculate p-value using those 3 independent experiments and showed the representative of them.

Reviewer #3 (Remarks to the Author):

In the revision the authors have answered my queries and concerns in a satisfactory manner except for their response to my original point 4 about DNA methylation. As requested, they performed bisulfite DNA methylation analysis on two individual VL30 copies (copy 21 and copy 38) before and after SETDB1 knockdown but, curiously, they do not show all the CpGs for these copies in Supplementary figure 7b. These copies contain many more CpG sites and all should be assayed. It appears from the CpG spacing in the figure that the sites shown are all upstream of the probable TATAA box, whereas, at least for one of the copies, there are 8 additional CpGs downstream of the TATAA box. In this reviewer's experience, methylation status of CpGs downstream of the TATAA box is most likely to correlate with transcriptional activity of LTRs. The LTRs are not so long so it should be not be a major technical challenge to amplify the whole LTR by placing one primer in the 5' interior of the VL30 and one in the unique flanking region. As it stands, the authors' conclusion "that there is no clear reduction of DNA methylation upon depletion of Setdb1, at least at these specific VL30 loci" cannot be made. This is an important point.

Response: Thank you for the reviewer's great comment. We have not paid attention about this TATAA box upstream/downstream issue. We looked back the location of TATTA box in VL30 5' LTR which we analyzed for the bisulfite sequencing. Although VL30 U3 class I 5' LTR does not have the typical "TATAA" sequence, there is "TAAAA" sequence where TATAA locates in other classes of VL30 LTR. And, we indeed examined CpG methylation downstream of this TAAAA site. To amplify unique sequence of #21 and #38 copies (both non-PBS-pro), we had to design relatively longer-size (1st)PCR (~700bp), but we could not amplify entire LTR and missed last 4 CG sites downstream of "TATAA" box. Thus, we examined 5/9 CpG sites downstream of TATTA in 5' LTR (or 5/10 CpG sites in entire 5' LTR). To avoid confusion, we added an additional supplementary figure to indicate the location of CG sites which we analyzed (Supplementary Fig. 8). In contrast to the analyzed VL30 PBS-pro LTR we amplified using Macfarlan's PCR primers (Wolf et al, *Genes Dev* 2015), the number of CG sites of #21 and #38 in LTR is relatively

less. Furthermore, we slightly changed our description of this result as follows,

“we did not detect clear reduction in DNA methylation at the analyzed VL30 5’ LTR sequences (individual locus of non-PBS-Pro copies, #21, #38 and pool of PBS-Pro elements loci) after the depletion of Setdb1 (Supplementary Fig. 7b and 8).”

REVIEWERS' COMMENTS:

Reviewer #3 (Remarks to the Author):

The authors have satisfactorily answered my remaining concern.